# Gut microbiome modulates *Drosophila* aggression through octopamine signaling

Yicong Jia[1,2,7], Shan Jin[1,7], Kunkun Hu[1,2], Lei Geng[3], Caihong Han[4], Ruxue Kang[2], Yuxin Pang[2], Erjun Ling[3], Eng King Tan[5], Yufeng Pan [4,6✉] & Wei Liu [1,2,4,5✉]

Gut microbiome profoundly affects many aspects of host physiology and behaviors. Here we report that gut microbiome modulates aggressive behaviors in *Drosophila*. We found that germ-free males showed substantial decrease in inter-male aggression, which could be rescued by microbial re-colonization. These germ-free males are not as competitive as wild-type males for mating with females, although they displayed regular levels of locomotor and courtship behaviors. We further found that *Drosophila* microbiome interacted with diet during a critical developmental period for the proper expression of octopamine and manifestation of aggression in adult males. These findings provide insights into how gut microbiome modulates specific host behaviors through interaction with diet during development.

[1] State Key Laboratory of Biocatalysis and Enzyme Engineering, Hubei Province Key Laboratory of Biotechnology of Chinese Traditional Medicine, National & Local Joint Engineering Research Center of High-throughput Drug Screening Technology, School of life science, Hubei University, Wuhan, China. [2] Department of Medical Laboratory Science, Fenyang College, Shanxi Medical University, Shanxi, China. [3] Key Laboratory of Insect Developmental and Evolutionary Biology, Institute of Plant Physiology and Ecology, Shanghai Institutes for Biological Sciences, Chinese Academy of Sciences, Shanghai, China. [4] The Key Laboratory of Developmental Genes and Human Disease, School of Life Science and Technology, Southeast University, Nanjing, China. [5] Department of Neurology, National Neuroscience Institute, Singapore General Hospital Campus, Singapore, Singapore. [6] Co-innovation Center of Neuroregeneration, Nantong University, Nantong, China. [7] These authors contributed equally: Yicong Jia, Shan Jin. ✉email: pany@seu.edu.cn; liuwei@sxmu.edu.cn

All metazoans harbor complex consortia of microbial species, collectively referred to as the microbiota or microbiome. The microbiome is widely acknowledged as a central regulator that modulates a range of host physiology and behaviors[1–4]. Mounting studies revealed that the microbiome affects neurodevelopment, cortical myelination, the function of the blood–brain barrier as well as complex behaviors, including locomotion, mating, anxiety, and cognition[5–11]. Alteration of the composition and function of microbiome is tightly associated with various neuropsychiatric disorders—such as social activity, stress, and anxiety-related responses in humans[12]. Nevertheless, it is largely unclear whether such microbial influences on behaviors stem from neurodevelopment during critical early-life periods or temporarily altered physiology during adulthood. Indeed, early-life microbial colonization promotes the maturation and homeostasis of the immune system through the release of microbial products[13,14]. Microbiome in early life also modulates sympathetic neurons via a gut–brain circuit, and regulates fear extinction learning[15,16]. A recent study elucidates the underlying mechanism by which bioactive metabolites produced by early-life microbiome can shape the metabolome of the nervous system and induces changes in specific neuronal circuits[17]. As a result, this forging symbiosis substantially imparts long-lasting effects on host fitness, and expands the survival and propagation capacity of hosts in the environment[13,18]. However, the precise mechanisms by which the microbiome influences neuronal development and host behaviors remain elusive. Due to the diversity of mammalian microbiome and the complexity of microbial metabolism, understanding and harnessing their potential is a challenging task. The fruit fly Drosophila melanogaster has a relatively simple commensal community[19], and fruitful genetic and neuronal tools for manipulating well-defined innate and learned behaviors[20], making it as a powerful and experimentally tractable system to study how microbiome modulates host behaviors.

Intermale aggression is an innate species-typical social behavior that is evolutionarily conserved across multiple species[21]. Expression of agonistic behavior is ordinarily observed between conspecific males in conflict over access to desired resources, including territory, food, and mating partners. D. melanogaster displays a gender-specific repertoire of stereotyped aggressive behaviors, which provides a feasible model for investigating the genetic and neural basis of aggression[22,23]. Although substantial studies have shown that aggression is shaped by genetic and environmental factors, it is observed that high heterogeneity of aggressive manifestation exists among individuals, and the root cause of this heterogeneity is relatively unclear, with interactions among hosts, microbiome and environment usually offered as possible explanations[24]. In the wild, Drosophila mainly feeds on decaying fruits that are inhabited by a myriad of microbes[25]. Due to their saprophagous foraging behavior, Drosophila ingests a variety of microbes potentially from either food resources or the surrounding environment from early development. Indeed, the establishment and maintenance of the microbiome mainly rely on the ingestion of environmental bacteria in Drosophila[26], which may partially explain the heterogeneity of aggressive manifestation among individual flies. Recent findings uncovered important roles of Drosophila microbiome in regulating a number of different behaviors, including mating and egg-laying preference, locomotion, and food-seeking behaviors[6,8,10,11,27–30]. However, whether gut microbiome plays a role in modulating aggressive behaviors and contributes to the heterogeneity of aggressive manifestation is unclear. A previous study reported that pathogenic Wolbachia impairs male aggressive behavior in Drosophila by downregulation of the octopamine (OA) biosynthesis pathway[31], but whether commensal microbiome could modulate host aggressive behavior is not known.

In this study, we generated germ-free (GF) flies and investigated a range of behaviors including aggression, and found that gut microbiome specifically affects aggressive but not locomotor or courtship behaviors. We further revealed that the gut microbiome interacted with diet during a critical developmental period and promoted OA biosynthesis and aggression. Thus, our findings demonstrate the critical roles of commensal microbiome and diet in shaping aggressive behaviors, and provide a robust paradigm to further study the gut–brain interaction using Drosophila as a model.

## Results

**_Drosophila_ microbiome modulates aggression.** To investigate whether the microbiome modulates Drosophila aggressive behaviors, we generated GF flies as illustrated in Fig. 1a. Axenia of GF flies was confirmed by plating the homogenates on nutrient agar plates (Fig. 1b) or performing 16 S ribosomal DNA (rDNA) PCR (Fig. 1b and Supplementary Figs. 1a, b). We raised all flies using a food medium with 10% of yeast to compensate for the developmental delay of GF flies as previously used (see below)[29]. We firstly assayed aggression in conventionally reared (CR) and GF males using an aggression assay as described[32] (Fig. 1c). Lunging frequency within 10 min after initiation (lunges per minute) and fighting latency (time to initiate lunging) were used to measure aggressive behaviors as previously used[33]. We found that depletion of microbiome profoundly dampened aggressive behaviors in Drosophila males. Compared to CR–CR pairs, GF–GF pairs exhibited a significant decrease in lunging frequency (Fig. 1d and Movie S1, S2), and a delay in initiating lunging (Fig. 1e). We also found that GF-CR pairs showed decreased lunging frequency compared to CR–CR pairs and increased fighting latency (Fig. 1d, e and Movie S3), which could be due to a lower aggressive drive in the GF male; alternatively, there may be pheromone changes from the GF male's cuticle that decreased the CR opponent's aggressive behaviors[27]. To further confirm the influence of the microbiome on aggression, we generated GF flies using two other protocols and assayed their aggressive behaviors. Indeed, we observed decreased lunging frequency and increased fighting latency in these GF males as compared to CR males (Supplementary Fig. 2a–f).

To further determine whether Drosophila microbiome is indeed responsible for promoting aggression, we carried out rescue experiments by colonization with consortia of microbes or defined individual microbes. Firstly, we inoculated GF embryos with mixed bacteria (MB) upon embryo sterilization (Fig. 1a). Bacterial recolonization was confirmed by plating fly homogenates on nutrient agar plates (Fig. 1b) and performing 16 S rDNA PCR (Supplementary Fig. 1a, b). We then tested aggression in recolonized MB males and found that they exhibited much higher aggression than GF males, comparable to that in CR males (Fig. 1f, g), demonstrating that microbiome-derived signals are indeed responsible for establishing a regular level of aggressive behavior in males. Because the rescued aggression of MB flies precluded any side effect of the sterilization process on host aggression, we mainly compared phenotypes between GF and MB flies in our later experiments. As we recorded aggression for up to 30 min, we tried to analyze aggression in a longer period. We found that GF males have reduced aggression in 20 min after initiation, compared to CR and MB males, and lunge numbers in the first 10 min are generally more than those in the second 10 min in all males, suggesting a decrease of aggression over time, regardless of CR, GF, or MB males (Supplementary Fig. 3a–f). Thus, lunge numbers in the first 10 min period after initiation well represent levels of aggression in males. Because it is very time-consuming for manual score of the lunging behavior, we mainly quantified lunges numbers within first 10 min after initiation in later experiments.

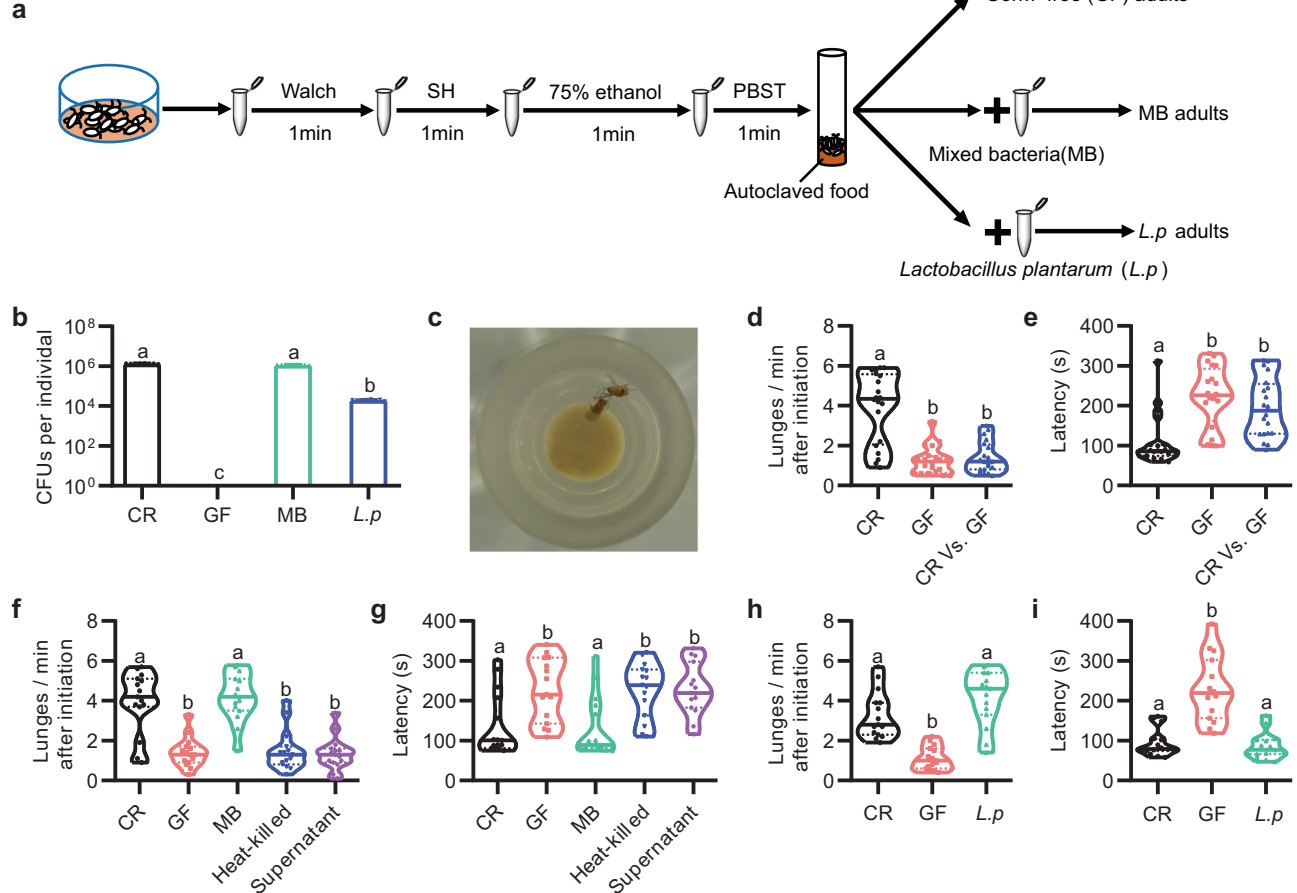

**Fig. 1 Gut microbiome modulates intermale aggression. a** Schematic illustration to generate germ-free (GF) and gnotobiotic flies. Embryos were successively sterilized with sanitizer walch, sodium hypochloride (SH), ethanol, and PBS containing 0.01% TritonX-100T (PBST). *Drosophila* indigenous bacteria and *Lactobacillus plantarum* (*L.p*) were inoculated to generate MB and *L.p*-associated adults. **b** Internal bacterial load of CR, GF, MB, and *L.p* adults. Bacterial load is illustrated as the colony forming units (CFU). $n = 8$ for each. Error bars indicate SEM. **c** A movie still that showed lunging behavior between two males. **d, e** Violin plots showing intermale lunging frequency (**d**) and latency (**e**) of CR–CR, GF–GF, and CR–GF pairs. $n = 20$ for each. **f, g** Effects of live mixed bacteria (MB), heat-killed MB, or MB supernatants on lunging frequency (**f**) and fighting latency (**g**) in GF males. $n = 15$ for each. **h, i** Lunging frequency (**h**) and fighting latency (**i**) were restored by *L.p* recolonization in GF males to the level of CR males. $n = 15$ for each. For all variables have different letters, they are significantly different ($p < 0.01$). Kruskal–Wallis test followed by Dunn's multiple comparisons test.

Compared to males, *Drosophila* females exhibit distinct fighting patterns that are characterized by head butting and swatting to their opponents[34]. To test whether the fly microbiome affects female aggression, we examined the frequency of head butting within 10 min after initiation and the latency to initiate head butting. We found that the frequency of head butting was significantly reduced, and the latency was significantly prolonged in GF–GF female pairs compared to CR–CR or MB–MB pairs (Supplementary Fig. 4a, b). Taken together, these results indicate that the microbiome promotes aggressive behaviors in both *Drosophila* males and females.

To test if live bacteria are required for the aggression promoting phenotype in males, we further inoculated heat-killed bacteria or supernatants containing bacterial metabolites to fly food with GF embryos, and found that neither condition could restore aggressive behaviors in males (Fig. 1f, g), indicating that it is live bacteria but not their metabolites responsible for promoting aggressive behaviors. To ensure that bacterial load approximates the one in the gnotobiotic experiments, the bacterial titers in fly food vials were assessed using nutrient agar plates. Bacteria reached a plateau at $2.6 \times 10^7$ CFUs/ml after 4 days of growth under this condition (Supplementary Fig. 5a). We added about tenfold of maximal amount of heat-killed

bacteria ($2.6 \times 10^8$ CFUs/ml) to GF flies. Our data showed that GF flies with high levels of dead bacterial biomass still displayed low level of aggression (Supplementary Fig. 5b, c). These results clearly indicate that bacteria must be metabolically active to stimulate aggressive behavior in *Drosophila* males.

We next analyzed compositions of above MB by using 16 S rDNA sequencing. In agreement with previous studies[5], our laboratory-reared flies were typically associated with ~30 bacterial species, dominated by the acetic acid bacteria (*Commensalibacter* and *Acetobacter*) and the lactic acid bacteria (*Lactobacillus*, *Streptococcus*, and *Enterococcus*) (Supplementary Fig. 6a, b). Moreover, principal component analysis showed that the composition of *Drosophila* microbiome was grouped away from bacterial community in fly diet (Supplementary Fig. 6c, d), indicating that host determinants contribute to microbial community composition. As we also observed a well-known *Drosophila* endosymbiotic bacterium, *Wolbachia*, we generated *Wolbachia*-free flies using tetracycline for three generations as previously described[35] (Supplementary Fig. 7a). We found that *Wolbachia*-free GF males still displayed decreased aggressive behaviors compared to CR and MB counterparts (*Wolbachia*-free) (Supplementary Fig. 7b–d), suggesting that *Wolbachia* associated with our flies are not responsible for the microbiome-mediated aggression. *Lactobacillus*

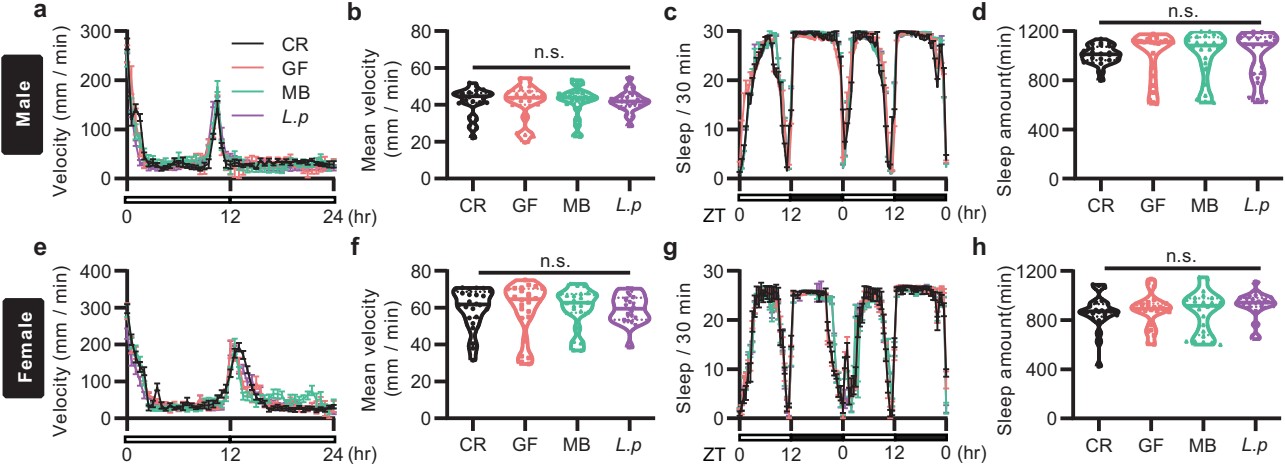

**Fig. 2 Gut microbiome does not affect locomotor behaviors. a**, **b** *Drosophila* 24-h walking speeds (**a**) and the average velocity (**b**) of GF males were not significantly different compared to CR, MB, and *L.p* counterparts. n = 24 for each. **c**, **d** GF males had largely unaltered sleep profiles (**c**) and total sleep amounts (**d**) in males compared to CR, MB, and *L.p* counterparts. n = 24 for each. **e**, **f** 24-h walking speeds (**e**) and the average velocity (**f**) of GF females were not significantly different compared to CR, MB, and *L.p* counterparts. n = 24 for each. **g**, **h** GF females had largely unaltered sleep profiles (**g**) and total sleep amounts (**h**) compared to CR, MB, and *L.p* counterparts. n = 24 for each. n.s. not significant. One-way ANOVA followed by Tukey's multiple comparisons test. For **a**, **c**, **e**, and **g**, Error bars indicate SEM.

*plantarum* (hereafter referred to as *L.p*), a commensal bacterium in *Drosophila* guts, is abundant in our sequencing data and found to be sufficient to recapitulate the natural microbiome growth-promoting effect[36]. We found that recolonization with *L.p* alone fully restored aggression levels in GF males (Fig. 1h, i). We also recolonized GF males with a few other bacterial species associated with *Drosophila*, and found that commensal bacteria, including *Acetobacter*, *Lactobacilli*, and *Enterococci*, promote aggressive behavior (Supplementary Fig. 7e, f). Intriguingly, this promotion of aggression was not observed when a pathogenic bacterium (ECC15) or a fungus (*Diaporthe* FY) was utilized. Together, these results reveal crucial roles of commensal bacteria in promoting aggressive behavior in males.

**Microbiome-mediated aggression is not due to locomotor defects.** Coordinated locomotor behavior is critical for social behaviors. It is conceivable that a general locomotor defect would decrease aggressive behaviors. Indeed, a previous study found that the microbiome suppressed locomotor such that GF flies were hyperactive[10], raising the question of whether microbiome would modulate aggressive behaviors through changes in locomotor activity. To test this possibility, we assayed walking speed of individual males for 24 h using video tracking as previously did[37]. Intriguingly, we did not find any significant difference of walking speed between CR, GF, and MB males (Fig. 2a, b). Additionally, *L.p*-associated flies displayed a comparable average walking speed (Fig. 2a, b). To further confirm these findings on locomotor behaviors, we assayed sleep of individual males using the *Drosophila* Activity Monitor (DAM2) as previously used[37], and did not observe any significant difference in sleep amounts in CR, GF, MB, and *L.p* males (Fig. 2c, d). We also tested walking speed of individual females for 24 h as well as their 2-day sleep, as female locomotor activity was mainly used in the previous study[10], and again found that average walking speeds (Fig. 2e, f) or sleep amounts (Fig. 2g, h) were not significantly different in CR, GF, MB, and *L.p* females. As we generated GF flies slightly different from the previous study that used irradiated fly food[10], we followed their assay generating GF males and females, and then tested the 24-h walking speed. We found no significant difference of the 24-h average walking speed in these GF flies compared with their CR controls (Supplementary Fig. 8a–f). The previous study

only analyzed walking speed for 10 min, while we analyzed average walking speed for 24 h. To assess alternative explanations for the discrepancy, we further analyzed the average walking speed every 10 min for 24 h (thus in total 144 time points). We indeed found a few cases out of 144 time points where walking speeds of CR and GF flies were significantly different (17 out of 144 time points for males, Supplementary Fig. 8c; 11 out of 144 time points for females, Supplementary Fig. 8f), but these differences are not consistent in one direction (e.g., higher or lower), and likely due to rapid changes of locomotion around morning and evening peaks for circadian regulation (see Discussion)[38]. As these differences were only inconsistently observed in short time scales (10 min), but not in a longer time scale, we conclude that *Drosophila* microbiome does not significantly affect locomotor behaviors in either sex, and its promoting role in aggression is not attributed to locomotor changes. Indeed, males of another wild-type strain Oregon-R have much lower level of aggression but comparable locomotor activity compared to Canton-S males (Supplementary Fig. 9a–c), further supporting the notion that locomotor and aggression are not directly correlated.

To test whether decreased aggression in GF flies could be due to potential increase of other behaviors, we tested feeding and grooming behaviors and found comparably low level of these behaviors in CR–CR, GF–GF, and MB–MB paired males (Supplementary Fig. 10a–c), suggesting that the decreased aggression in GF males is not attributed to competitions with other exclusive behaviors. Thus, these findings indicate that gut microbiome specifically modulates aggressive behaviors in *Drosophila*.

**GF males are less competitive in mating success.** Given that success in aggressive interactions is reciprocally correlated with fitness for mating, we further explored whether *Drosophila* microbiome affects male courtship behaviors by investigating courtship behaviors of GF and MB males towards wild-type virgin females. We found that GF and MB males courted virgin females in similarly high levels (Fig. 3a), and their copulation rates were also comparable (Fig. 3b), which is consistent with a previous study[11].

It has been well established that aggression plays crucial roles in courtship competition[22,39,40]. Given lower levels of aggression in GF males, it is conceivable that GF males might be less

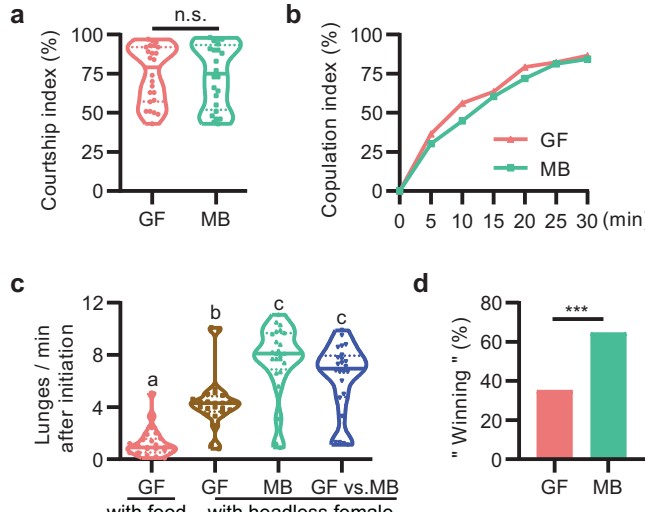

**Fig. 3 Germ-free males have normal courtship but are less competitive. a**, **b** GF males did not exhibit a significant difference in courtship index (**a**) or copulation index (**b**) compared to MB ones. Conventionally reared wild-type virgin females were used as targets. $n = 24$ for courtship, $n = 96$ for copulation. n.s. not significant. Two-tailed Mann–Whitney $U$-test. **c** Lunging frequency of GF–GF, MB–MB, and GF–MB male pairs were significantly enhanced in the presence of a headless virgin female. $n = 24$ for each. For all variables have different letters, they are significantly different ($p < 0.05$). Kruskal–Wallis test followed by Dunn's multiple comparisons test. **d** Percentage of successful copulation by GF or MB males with females in a competitive courtship assay. $n = 85$. ***$p < 0.001$, chi-square test.

competitive in copulation success than MB ones. To test this possibility, we firstly tested intermale aggression among GF–GF, MB–MB, and GF–MB pairs in the case of a decapitated virgin female, and found consistently enhanced lunging frequency in both GF–GF and MB–MB pairs (Fig. 3c), suggesting that enhanced mating driving could promote intermale aggression in both GF and MB males. Remarkably, a headless female also increased lunging frequency in GF-MB pairs to a level comparable in MB–MB pairs (Fig. 3c). To test whether GF and MB males are equally competitive for copulation, we assessed copulation success within 30 min by placing a GF male and a MB male with different color dots on their thorax with an intact virgin female, and found that MB males had significantly more chances to gain copulation with females compared to GF males (Fig. 3d, 55 out of 85 won, and Movie S4). Taken together, these results indicate that GF males are less competitive to gain successful copulation with females despite regular courtship drive.

**Microbiome promotes aggression through OA signaling.** Extensive studies have demonstrated that OA plays a crucial role in aggressive behaviors in *Drosophila* and other insects[32,41–43]. It was recently reported that the microbiome decreased OA levels in *Drosophila* females[10]. To test whether the microbiome could modulate aggressive behaviors by altering OA expression levels, we firstly measured relative mRNA levels of *tyrosine decarboxylase 2* (*Tdc2*) and *tyramine beta-hydroxylase* (*Tβh*) encoding the enzymes for OA synthesis as well as other aggression-associated genes. We found that the mRNA level of *Tdc2* in heads of GF males was significantly lower than that in MB ones (Fig. 4a). On the other hand, mRNA levels of other tested genes including *Tβh*, *TH*, *Tk*, *Trh*, and *Gad1* were comparable in heads of GF and MB males (Fig. 4a). Furthermore, the mRNA level of *Tdc2* but no other examined genes was significantly decreased in heads of GF females compared with that in MB females (Fig. 4b). These results

indicate that *Drosophila* microbiome specifically elevates OA signaling to stimulate aggressive behaviors. As we found an opposite effect of microbiome on modulating OA levels to a previous study[10], we further measured the total level of OA in brains of GF and MB males with high-performance liquid chromatography (HPLC). The HPLC result showed that the level of OA in heads of GF males was only ~27% of that in MB ones (average OA concentration per head: $1.19 \pm 0.18$ pg for GF males and $4.42 \pm 1.6$ pg for MB males, Fig. 4c, d). To further confirm expression change of *Tdc2* as we described above, we stained brains of CR, GF, and MB males with anti-Tdc2 antibody. We found that the immunoreactivity to Tdc2 was grossly reduced in GF males compared to CR and MB males (Fig. 4e, f). We then divided these Tdc2-positive neurons into five distinct clusters based on previous classifications[44,45]. We found that anti-Tdc2 signals in three out of five subsets of Tdc2 neurons in GF males are significantly decreased compared with that of CR and MB counterparts (Fig. 4e, g). These results indicate that OA production is decreased in subsets of octopaminergic neurons in males depleted of microbiome.

Since we found that OA levels were much lower in GF males, we assumed that feeding GF males with the OA agonist chlordimeform (CDM) would restore the impaired aggression in GF males. Indeed, we found that supplementation of CDM with a concentration of 0.1 mg/ml fully restored aggressive levels in GF males to ones in MB males (Fig. 4h). These results strongly indicate that the microbiome positively regulates octopaminergic pathways to modulate *Drosophila* aggressive behaviors. To further corroborate the role of OA in the reduction of aggression in GF males, we sought to activate OA-expressing neurons by expressing the cation channel dTrpA1[46] in *Tdc2-GAL4* neurons in GF males. We found that acutely activating octopaminergic neurons in GF males at 29 °C significantly elevated intermale aggression compared with genetic controls (Fig. 4i). Furthermore, chronic activation of octopaminergic neurons by expressing NaChBac, a voltage-sensitive sodium channel derived from bacteria[47], also significantly increased intermale aggression in GF males compared with genetic controls (Fig. 4j). Axenia of each fly line was verified by performing 16 S rDNA PCR (Supplementary Fig. 11a, b). Together these results indicate that OA signaling in octopaminergic neurons is responsible for microbiome-mediated changes of aggressive behaviors.

**Microbiome promotes aggression and OA production during development.** Emerging studies showed that early-life exposure to microbiome promotes the maturation and homeostasis of the organism systems through the release of microbial products, imparting long-lasting effects on host physiology and behaviors[5,29,36]. These results prompted us to explore whether there was a crucial developmental stage when *Drosophila* microbiome is required for promoting aggressive behaviors. Thus, we set out to inoculate GF embryos with MB at different developmental time points (Fig. 5a). We found that adding MB 36 h after egg laying (AEL) fully restored aggressive behaviors into a level comparable to CR males, while adding MB 48, 72, or 96 h AEL only partially restored aggressive levels in GF males, and adding MB during adulthood did not promote aggression in GF males at all (Fig. 5b). Consistent with these findings, the level of *Tdc2* mRNA in GF males that were inoculated with bacteria 96 h AEL was significantly lower compared to MB males that were inoculated with bacteria immediately after the sterilization process (Fig. 5c). These results suggest that the manifestation of aggression in adult males requires *Drosophila* microbiome during a critical developmental period (48–96 h AEL). We further examine whether constant association between *Drosophila* and microbiome is necessary to sustain microbiome-mediated

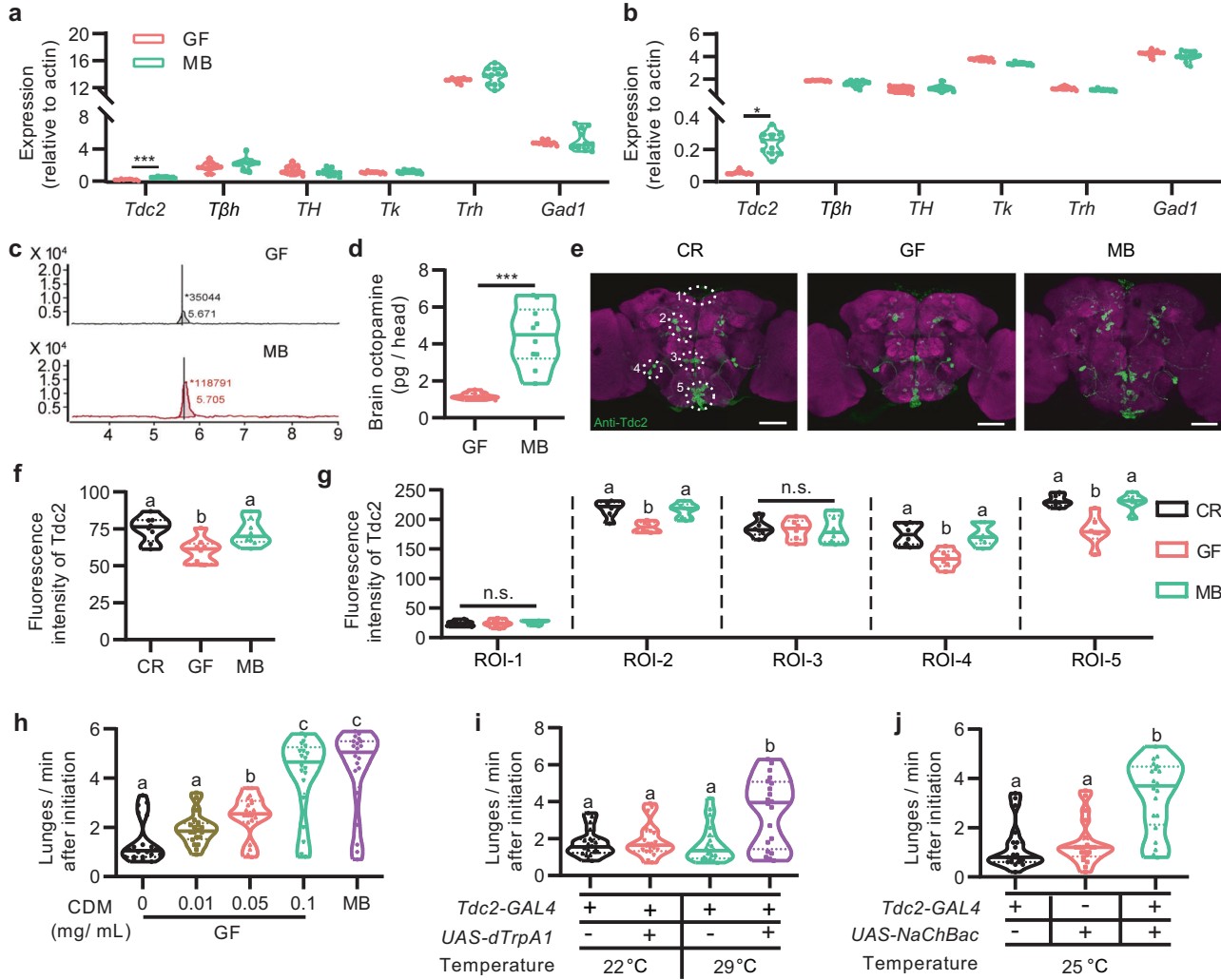

**Fig. 4 Gut microbiome promotes aggression through octopamine signaling. a**, **b** Relative expression of *Tdc2*, *TβH*, *TH*, *Tk*, *Trh*, and *Gad1* in brains of males (**a**) and females (**b**). *n* = 12 based on three replicates for each. **p* < 0.05, ****p* < 0.001, Two-tailed Mann–Whitney *U*-test. **c**, **d** The level of octopamine in GF male brains was significantly lower than that in MB males. *n* = 10 for each. ****p* < 0.001, Two-tailed Mann–Whitney *U*-test. **e** Representative images of the anti-Tdc2 signals in adult brains. Tdc2-expressing neurons were divided into five distinct clusters based on their location. Scale bars, 50 μm. **f**, **g** The average fluorescence intensity was calculated in total brain region (**f**) and five regions of interest (**g**). *n* = 8 for each. **h** OA agonist CDM elevated lunging frequency of GF in a dosage-dependent manner. *n* = 20 for each. **i**, **j** Activating octopaminergic neurons with dTrpA1 (**i**) or NaChBac (**j**) significantly increased lunging frequency of GF males. *n* = 20 for each. For all variables have different letters, they are significantly different (*p* < 0.05). Kruskal–Wallis test followed by Dunn's multiple comparisons test.

aggression promotion by transferring CR larva to food with a cocktail of antibiotics at different developmental time points (Fig. 5d). We found that depleting bacteria 24–72 h AEL significantly decreased male aggression, while bacteria removal 96 h AEL did not affect male aggression (Fig. 5e). Accordingly, the level of *Tdc2* mRNA decreased significantly in males with microbiome depleted 24–72 h AEL, but unaffected in males with microbiome depleted 96 h AEL (Fig. 5f). Together these results indicate that microbiome is required during 48–96 h AEL, but dispensable after this specific period, for promoting OA production and aggression in adult males.

Since microbiome functions early during development to promote aggression through the OA signaling, we further asked whether the OA signaling would be affected during the critical developmental period. Thus, we compared *Tdc2* expression levels using the anti-Tdc2 antibody in CNS of larva 36–48 h AEL as well as larva 72–84 h AEL in CR, GF, and MB males. We divided Tdc2 expression into three subsets based on their location (Fig. 5g, j), and found that numbers of *Tdc2*-expressing neurons in larva

neither 36–48 h AEL nor 72–84 h AEL were grossly affected by microbial depletion (Fig. 5h, k). However, Tdc2 expression was significantly decreased in two subsets of *Tdc2*-expressing neurons of GF larva 72–84 h AEL (Fig. 5i), but unaffected in GF larva 36–48 h AEL (Fig. 5l), suggesting that the microbiome-mediated increase in OA signaling originates approximately from 72–84 h AEL. These results further indicate that the microbiome is required during a critical developmental period to enhance OA production and promote male aggression.

**Microbiome synergizes with diet to promote male aggression.** Previous studies reported that *Drosophila* microbiome sustains optimal larval development upon nutrient scarcity[5,36]. Consistent with these reports, we found that GF flies underwent a prolonged developmental process under lower nutrition condition (0.5% yeast), but a diet with yeast enrichment (>2.5%) was sufficient to restore this developmental deficit of GF flies (Fig. 6a). As yeast provides proteins as well as most other noncaloric nutrients for fly development in laboratory, and GF flies have prolonged

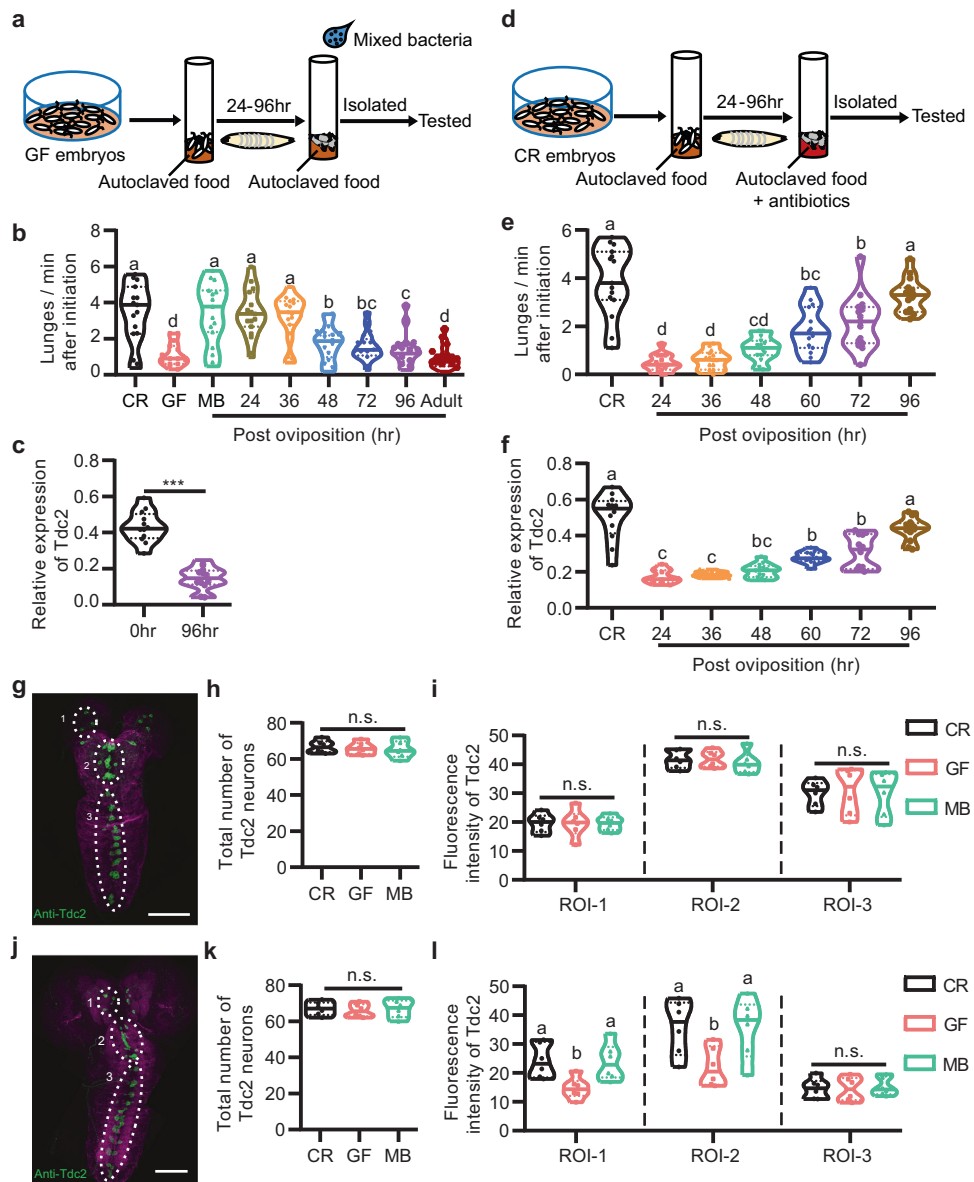

**Fig. 5 Microbiome promotes aggression during a critical developmental period. a** Experimental setup to assess the impact of delayed association of microbiome on aggression gain. **b** Lunging frequency of GF males recolonized at different developmental time points. Recolonization of mixed bacteria in early stages of development, but not later or adult stage, fully or partially restored aggression in GF males. $n = 15$ for each. **c** Relative expression of *Tdc2* mRNA in GF males recolonized with mixed bacteria at different developmental stages. $n = 12$ based on three replicates for each. ***$p < 0.001$, Two-tailed Mann–Whitney *U*-test. **d** Experimental setup to assess the impact of bacterial ablation during development on aggression of adult males. **e** Lunging frequency of adult males after above manipulation. Efficient bacterial ablation by antibiotics was assessed by plating adult homogenates at the time of collection on LB agar plates. $n = 15$ for each. **f** Relative expression of *Tdc2* mRNA in adult males after above manipulation. $n = 12$ based on three replicates for each. **g** Representative image of the anti-Tdc2 signals in CNS of larva 36–48 h AEL. Scale bars, 50 μm. **h** Total numbers of Tdc2 neurons in CR, GF, and MB larva 36–48 h AEL were not significantly different. $n = 6$ for each. **i** Fluorescence intensities of anti-Tdc2 signals in ROI-1, ROI-2, and ROI-3 were calculated in CR, GF, and MB larva 36–48 h AEL. $n = 6$ for each. **j** Representative image of the anti-Tdc2 signals in CNS of larva 72–84 h AEL. Scale bars, 50 μm. **k** Total numbers of Tdc2 neurons in CR, GF, and MB larva 72–84 h AEL were not significantly different. $n = 6$ for each. **l** Average fluorescence intensities of anti-Tdc2 signals in ROI-1, ROI-2, and ROI-3 were calculated in CR, GF, and MB larva 72–84 h AEL. $n = 6$ for each. For all variables have different letters, they are significantly different ($p < 0.05$). If two variables share a letter, they are not significantly different ($p > 0.05$). Kruskal–Wallis test followed by Dunn's multiple comparisons test.

developmental process with lower yeast feeding, we hypothesized that the microbiome might modulate aggression in a diet/nutrition-dependent manner. To test this hypothesis, we assayed intermale aggression in CR, GF, and MB males raised in food with different concentrations of yeast, and found that feeding on food with higher yeast concentration promoted intermale aggression in CR and MB males (Fig. 6b, c), suggesting that

nutrition is another major factor that affects aggressive behaviors in *Drosophila* males. Interestingly, GF males displayed comparable low level of aggression no matter raised on low-yeast or high-yeast diet (Fig. 6b, c). To test if higher concentrations of yeast could compensate for the reduction of aggression, we raised GF flies on a diet with 15 or 20% yeast, but found that neither 15% nor 20% yeast was able to restore the impaired aggression in GF

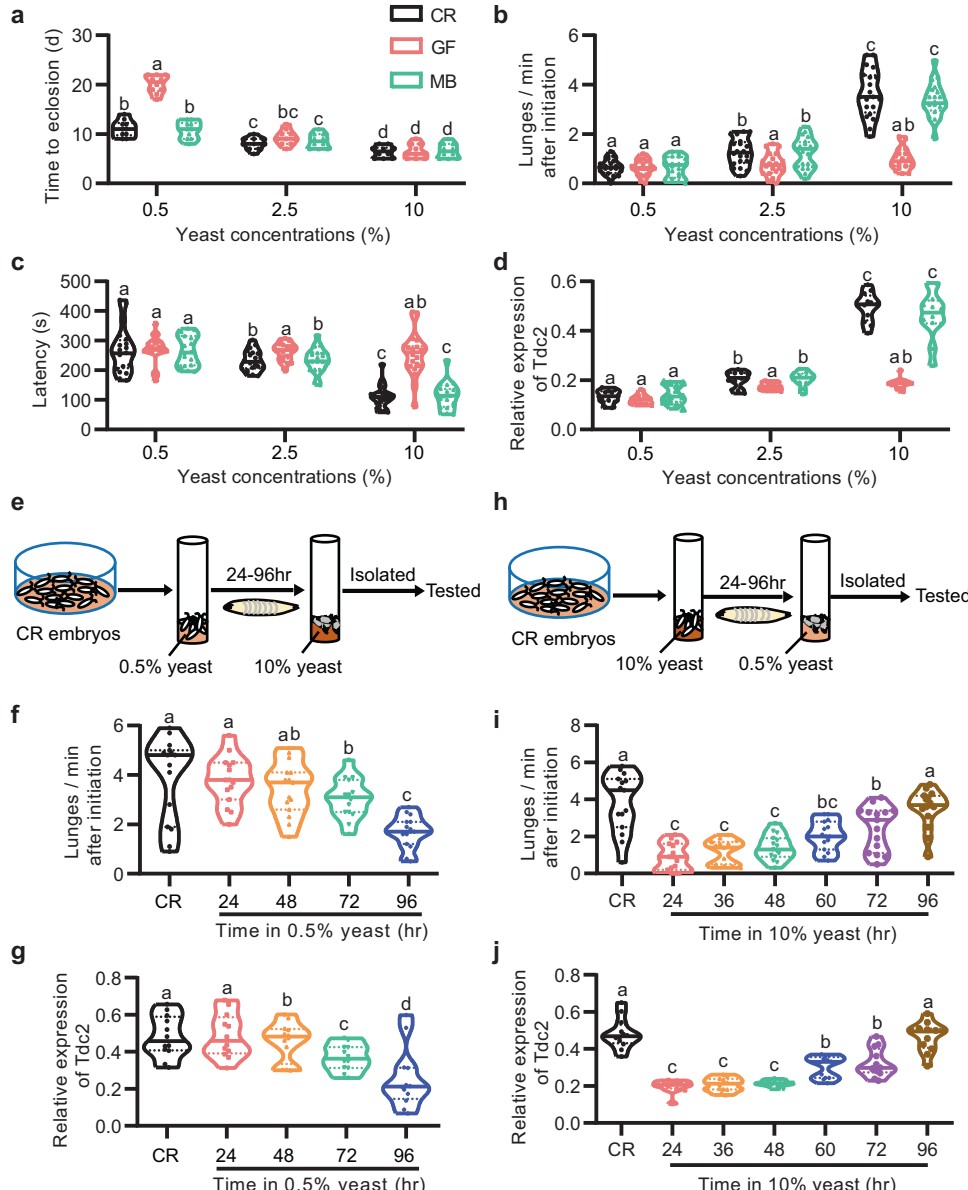

**Fig. 6 Microbiome interacts with diet to modulate aggressive behaviors. a** The developmental time to eclosion of CR, GF, and MB flies raised in food medium with different yeast concentrations (0.5, 2.5, and 10%). $n = 10$ vials for each. **b, c** Lunging frequency (**b**) and fighting latency (**c**) of CR-CR, GF-GF, and MB-MB males raised in food medium with different yeast concentrations. $n = 20$ for each. **d** Relative expression of *Tdc2* mRNA in CR, GF, and MB flies raised in food medium with different yeast concentrations. $n = 12$ based on three replicates for each. **e** Experimental setup to assess the impact of nutrition on aggression by transferring larvae from poor nutrition food to rich one at different developmental stages. **f** Lunging frequency of adult males after above manipulation. $n = 15$ for each. **g** Relative expression of *Tdc2* mRNA in adult males after above manipulation. $n = 12$ for each. **h** Experimental setup to assess the impact of nutrition on aggression by transferring larvae from rich nutrition food to poor one at different developmental stages. **i** Lunging frequency of adult males after above manipulation. $n = 15$ for each. **j** Relative expression of *Tdc2* mRNA in adult males after above manipulation. $n = 12$ based on three replicates for each. For all variables have different letters, they are significantly different ($p < 0.05$). If two variables share a letter, they are not significantly different ($p > 0.05$). Kruskal–Wallis test followed by Dunn's multiple comparisons test.

flies (Supplementary Fig. 12), indicating that microbiome cannot be substituted with rich nutrition during development to promote aggression. Accordingly, the levels of *Tdc2* mRNA were significantly elevated in CR and MB adult brains if raised throughout development with higher yeast feeding, while levels of *Tdc2* mRNA in GF males were equally low under feeding with different yeast concentrations (0.5, 2.5, and 10%), and lower than those in CR and MB males raised on food with 2.5 or 10% yeast (Fig. 6d). These results indicate that the microbiome promotes male aggression in a diet-dependent manner, and also revealed an important role of diet during development for establishing a

regular level of aggression and aggression-associated OA signaling in adult males.

We next asked whether undernutrition only during the abovementioned developmental period would abolish the effect of the microbiome on aggression. We reared CR embryos in food with poor nutrition (0.5% yeast) and switched them to a diet with rich nutrition (10% yeast) 24–96 h AEL (Fig. 6e). We found that adult males, if raised with 0.5% yeast for more than 48 h AEL, exhibited a significant decrease in aggression, as well as in *Tdc2* expression (Fig. 6f, g), and the effect is stronger if raised with 0.5% yeast for longer time (Fig. 6f, g). These results indicate that

nutrition scarcity stunts the promotion of microbiome-mediated aggression and OA production. We also tested whether under-nutrition during or after the critical developmental period would impair male aggression. We reared CR embryos in food with 10% yeast and switched them to a diet with 0.5% yeast 24–96 h AEL (Fig. 6h). We found that males showed decreased aggression and *Tdc2* expression if transferred to 0.5% yeast food 24–72 h AEL, but regular aggression and *Tdc2* expression if transferred to 0.5% yeast food 96 h AEL (Fig. 6i, j). These results indicate that nutrition is necessary to sustain microbiome-mediated aggression of males during the critical developmental period.

## Discussion

Accumulating evidence indicates that the microbiome affects a broad spectrum of animal physiology and behaviors[1,48,49]. It remains unclear whether/how *Drosophila* microbiome is required to modulate innate and social behaviors, including locomotion, courtship, and aggression despite of much advanced genetic tools in this animal model[20]. In this study, we demonstrated that microbiome specifically modulates aggressive behaviors in both males and females. Aggression in *Drosophila* has long been considered to play a critical role in mate selection[50]. Indeed, our findings indicate that GF males are less competitive in copulation with females compared with control CR males, indicating that microbiome depletion impairs optimal sexual fitness of adult males. Recolonization of MB, or specific commensal bacteria species including *Acetobacter*, *Lactobacilli*, and *Enterococci*, equally restored aggressive behaviors in GF males, suggesting that there are common genetic determinants in these bacteria species that promote adult aggression. Identifying these genetic determinants would help to understand in molecular details how microbiome modulates host behavior and enhances sexual fitness.

We identified the OA signaling responsible for the microbiome-mediated promotion of aggression. We found that microbiome-depletion resulted in a lower level of *Tdc2* expression in both GF male and female brains with qPCR. We further showed that there was a 73% reduction of OA level in GF male brains using the HPLC assay. Our finding that fly microbiome promotes OA production is generally consistent with previous findings in mammals that indigenous bacteria potentially impact behaviors by boosting biosynthesis of biogenic amines (e.g., serotonin)[51]. A previous study reported that pathogenic *Wolbachia* impairs male aggressive behavior by downregulation of the OA biosynthesis pathway, suggesting that pathogenic and commensal bacteria function oppositely in regulating host OA production and aggressive behaviors[31].

Although depletion of microbiome decreased OA signaling and substantially impaired aggression in both male and female flies, it did not significantly affect other behaviors. Our results are generally consistent with a previous finding that locomotor behaviors, sleep, and courtship behaviors in GF males are not virtually affected by the microbiome[11]. In contrast, Schretter et al. recently reported that depletion of microbiome increased OA signaling and induced hyperactivity in *Drosophila* females[10]. We suspected that such discrepancy might be due to different axenic culture conditions. However, we still did not find any significant difference in locomotion among CR, GF, and MB flies following their protocol for generating GF flies[10]. As we measured average walking speeds for 24 h, instead of 10 min as used by the previous study, we reanalyzed walking speeds every 10 min for 24 h, and observed locomotion differences in a few time points, but these differences were not consistently higher or lower in GF flies (thus not exist in a longer time scale), which may be due to large variation of locomotion during morning and evening peaks for circadian regulation[38]. Indeed, we note that Schretter et al. tested

walking speed of flies between ZT0 and ZT3 (lights are turned on at ZT0 and turned off at ZT12); however, locomotor behaviors vary a lot during this period after morning peak (it peaks at ZT0, and deceases by ~80% at ZT3 [from ~200 to ~30 mm/min], see Fig. 2a, e and Supplementary Fig. 8c, f), which may contribute to their observed locomotor differences. For such reason, we suggest future test of locomotor behaviors in a shorter time window (e.g., 30 min test between ZT1 and ZT2 with control and experimental flies tested simultaneously), or for a longer time (e.g., average walking speed for 3 h or even 24 h as used in this study). Another possible factor that may result in the discrepancy is that we used wild-type *Canton-S* (*wtcs*) flies, instead of *Oregon-R* flies as mainly used by Schretter et al., as *wtcs* were widely used for most behavioral tests and *Oregon-R* flies rarely displayed aggressive behaviors in our tests, although they have comparable locomotor activity with *Canton-S* flies. Regarding to why microbiome only affects aggression but not locomotor or other behaviors, there are at least two possibilities. One is that the OA level is not significantly reduced in specific neurons that may be responsible for a particular behavior. Alternatively, OA reduction in certain neurons is not sufficient to induce a behavioral change, e.g., even *tβh* or *tdc2* mutant flies showed comparable locomotor levels[32,52] although they are defect in starvation-induced hyperactivity[52].

A prominent feature of the role of microbiome on development and behavior is its dependence on diet. Firstly, GF flies have prolonged developmental process if raised with low level of yeast (0.5%), but develop normally with higher level of yeast (2.5 or 10%), consistent with previous studies[5,36,53]. Secondly, GF males showed reduced aggression only if raised with higher level of yeast (2.5 or 10%), as CR, GF, and MB males raised with low level of yeast (0.5%) all showed few and indistinguishable aggression. Furthermore, while CR males have much higher expression of *Tdc2* if raised with high level of yeast, GF males have similarly low level of *Tdc2* expression. Interestingly, supplement of rich yeast is only required during a critical developmental period, roughly 48–96 h AEL, for microbiome-mediated promotion of OA production and aggression. Thus, gut microbiome and a proper level of yeast consumption during a critical developmental period jointly promote OA production and aggressive behaviors in flies. Previous studies already showed that nutritional environment is a key factor involved in the microbiome-mediated development, metabolism, immunity, and behaviors[5,7,29,36,53,54]. Our results are generally consistent with these findings and further reveal that gut microbiome and diet interact to modulate neurotransmitter signaling and aggressive behaviors. It has been increasingly accepted that gut microbiome relies on diet to generate neuromodulators, provides missing nutrients to hosts, and modifies the availability of specific nutrients derived from the diet, consequently shaping the nutritional environment of hosts[55]. There are at least two underlying mechanisms by which the microbiome changes host physiology and behaviors by absorbing specific nutrients from the diet or de novo synthesizing special nutrients including neuroactive metabolites, amino acids, and short chain fatty acids[56,57]. It is plausible that microbiome may affect absorption/biosynthesis of the precursor of OA (Tyrosine), in addition to expression change of *Tdc2*, or leading to *Tdc2* expression change as a result of adaptation. Future studies on microbiome-induced alterations in the metabolome of *Drosophila* nervous system would improve the knowledge of microbe–nutrition–aggression interactions.

The finding that gut microbiome modulates aggressive behaviors raises a few questions. First, since recolonization of a few commensal bacteria fully restored aggression in GF males, identification of specific bacterial genes involved in OA production, and how they may interact with nutrition environment, are needed to further understand how gut bacteria modulate

aggression. Second, it is unclear if there are commensal bacteria that could oppositely modulate aggression. Future studies identifying commensal bacteria that positively or negatively modulate aggression would deepen our understanding and have potential implications utilizing commensal bacteria to modulate aggressive behaviors. Recently, it was reported that the microbiome correlated with conspecific aggression in a small population of dogs[58], highlighting that the microbiome may be useful for diagnosing aggressive behaviors prior to their manifestation and potentially discerning cryptic etiologies of aggression. Third, that microbiome synergizes with diet to promote aggressive but not other innate behaviors in our study is intriguing, especially given that *Drosophila* in the wild may be challenged with scare, dynamic, and highly diverse diets. Our results suggest that males fed on rich nutrition during development, with many commensal bacteria in their guts, have an advantage of reproduction. This association of microbiome and aggression thus is beneficial and selected, favoring the hologenome theory of evolution[59]. Our study using *Drosophila* thus provides a feasible model for elucidating the mechanism of how microbiome and diet interact to modulate biosynthesis of signaling molecules and host behaviors.

## Methods

**Fly culture and stocks**. Flies were reared and maintained at 25 ˚C, 60% humidity with a 12 h: 12 h light-dark circle unless otherwise noted. All flies used in this study were out-crossed to the wild-type *Canton-S* (*wtcs*) background and maintained on standard cornmeal agar medium except for the irradiation assay with antibiotics (see below). The standard food recipe was as follows: 105 g dextrose, 7.5 g agar, 26 g yeast, 50 g cornmeal, and 1 L purified $H_2O$ were mixed and boiled for 30 min with constant agitation, and 1.7 g Tegosept (Sigma Aldrich, St. Louis, MO, USA) dissolved in 8.5 ml 95% ethanol and 1.9 ml propionic acid (99%, Mallinckrodt Baker) was added. *wtcs*, *UAS-NaChBac*, and *Tdc2-Gal4* flies were gifts from Dr. Yi Rao in Peking University. *UAS-dTrpA1* and *UAS-Kir2.1* transgenic flies were from the Bloomington *Drosophila* Stock Center, and out-crossed to the *wtcs* background.

**Generation of GF flies**. GF flies were generated as described[30]. In brief, embryos were collected on agar media with grape juice within 8 h, rinsed with ddH₂O, and transferred into 1.5 ml Eppendorf tubes. Diluted sanitizer walch (1:30, Procter & Gamble Co., Cincinnati, OH, USA), 2.5% sodium hypochloride (Sigma Aldrich, St. Louis, MO, USA), 75% ethanol, and sterile PBS containing 0.01% TritonX-100T were used to sterilize embryos successively. These embryos were transferred into axenic food vials. Axenia of GF flies was confirmed by performing 16 S rDNA PCR (8 F: 5′-AGAGTTTGATCMTGGCTCAG-3′, 1492 R: 5′-GGMTACCTTGTTAC-GACTT-3′) on homogenates of the adult flies or by plating the homogenates on nutrient agar plates (peptone 10 g/L; beef extract powder 3 g/L; NaCl 5 g/L; agar 15 g/L). For axenic conditions, plastic vials containing fly food were autoclaved for 20 min. MB by grinding CR adult flies or *L.p* were inoculated to sterile food with GF embryos to generate bacteria recolonized flies (see Fig. 1a).

We also generated GF flies using two other protocols (Supplementary Fig. 2). One is to raise GF embryos that were obtained as described in Supplementary Fig. 2a in irradiated food medium. The other is to raise CR flies in irradiated medium with antibiotics (500 µg/ml ampicillin, Putney; 50 µg/ml tetracycline, Sigma; 200 µg/ml rifamycin, Sigma) as previously used[10], and described in Supplementary Fig. 2d.

**Bacteria culture and CFU counting**. MB were prepared by grinding 30 CR adult flies in 500 µL sterile phosphate buffered saline (PBS) with pestles as described[60], and the strain of *L.p* (*Lactobacillus plantarum*) with the Genbank accession number KY038178 was used. MB and *L.p* were cultured in LB broth and MRS broth medium at 30 ℃ respectively. Bacteria cells were harvested by centrifugation (3000×*g*, 5 min), washed twice in PBS, and resuspended in PBS to obtain $10^8$ cells/ml (OD₅₉₅ = 1). The bacterial suspension (0.1 OD or ~$10^7$ cells) was supplemented to autoclaved fly food with GF embryos to generate gnotobiotic flies. All the material to manipulate bacteria was sterilized prior to use.

To quantify the number of associated bacteria, 20 adult flies were surface sterilized by 70% ethanol for 2 min, and were homogenized with a pestle in 1.5 ml tube with 200 µL of sterile PBS. Bacterial load was calculated by plating tenfold serial dilutions of the homogenates on LB or MRS agar plates and incubating the plates at 30 ℃ for 48 h, and the numbers of CFU were counted.

**Treatment with heat-killed bacteria**. Treatment with heat-killed bacteria was performed with modification as previously described[61,62]. Briefly, the bacterial load (CFUs) in vials was assessed following bacterial inoculation in three independent experiments. To generate vials containing dead bacterial biomass, MB were prepared as described above. A number of cells corresponding to ten times the maximum number of bacteria measured in fly vials was harvested by centrifugation. For heat kill of bacteria, washed cells of MB were resuspended in 150 µl of 1x PBS and incubated for 4 h at 65 ℃. Once at room temperature, the cell suspension was added to vials with GF embryos.

**Aggression assays**. Males and females were collected shortly after eclosion and individually placed in tubes (5 ml tubes, Falcon) containing 0.5 ml of food. Aggression assays were carried out in fighting chamber at 25 ˚C, 60% humidity condition in the morning (from ZT0 to ZT3; lights are turned on at ZT0 and turned off at ZT12; ZT, Zeitgeber time) when the adults aged 5 to 7 days. The chamber was circular as shown in Fig. 1c. Two flies were transferred into the fighting chamber by gentle aspiration. The 20 mm × 20 mm glass coverslip was then rapidly covered the chamber. Fights were recorded for 30 min using a video camera (Sony FDR-AX40). We manually counted numbers of lunging in males and head butting in females. For intermale aggression, lunging frequency and latency of fighting were used to compare aggression levels. The latency of fighting was calculated by the time from the beginning of video recording to the first lunging, while the lunging frequency was calculated as the number of lunging per minute after lunging initiation. Lunging frequency was analyzed for a total of 20 min in Supplementary Fig. 3 and 10 min for all other results. For aggression tests between GF and CR (Fig. 1d, e) males, flies were discriminated by painting a small blue or red dot on the thorax of flies. Painted flies anaesthetized under light CO₂ were allowed to recover for 24 h before aggression assays.

Grooming behaviors during aggression were manually analyzed using video records of aggression. Grooming in our algorithm is defined as fly legs rubbing against each other or sweeping over the surface of the body and wings. The percentage of time used for grooming was calculated as previously described[63].

**Locomotion assays**. Locomotion was assayed and analyzed as previously described[64]. In brief, 4–6 days old group-housed flies (males or females) were anaesthetized on ice and introduced individually into round food chambers (20 mm diameter and 3 mm height) around 8 a.m., and allowed to recover for about 1 h at 25 ℃. Locomotor behavior was recorded by video camera starting from ZT0 for 24 h under constant light condition. The average walking velocity during the 24-h recording was quantified using the ZebraLab software system (ViewPoint Life Sciences, Montreal, Quebec, Canada).

**Sleep test**. Sleep was analyzed as previously described[64]. Individual males and females (3–5 days old) were placed in *Drosophila* activity monitor tubes (DAM2, TriKinetics Inc.) with fly food, and were entrained at 25 ˚C under 12 h: 12 h light: dark conditions for at least 2 days before sleep test. Sleep data were analyzed using DAMFileScan 113 and Matlab R2018a.

**Feeding assays**. Feeding assays were performed using 5–7 days old adult males. Satiated or 24-h starved males were transferred to fly food containing 0.5% w/v blue food dye (Erioglaucine Disodium Salt, Sigma). After 30-min exposure to blue dyed food, flies were homogenized in 0.5 ml PBS buffer with 1% Triton X-100 in Eppendorf tubes, and centrifuged (14,000x*g*) for 30 min to remove the debris. The absorbance of the supernatant was measured at 625 nm (Hitachi U-2001 Spectrophotometer). Age-matched flies in non-dyed food were utilized as the baseline during spectrophotometry. The amount of labeled food in the fly was calculated from a standard curve by serial dilution in water of blue food.

**Courtship assays**. Male flies were collected shortly after eclosion and housed in isolation for 4–6 days before courtship test. CR wild-type virgin females (5–7 days old) were used in all mating assays. The assay was performed at 25 ˚C, 60% humidity condition in the morning (from ZT0 to ZT3; lights are turned on at ZT0 and turned off at ZT12). The male–female courtship assay was carried out in a cylindrical two-layer courtship chamber, 10 mm (diameter) × 3 mm (height of each layer). Courtship behavior was recorded by camera for 30 min after a virgin female and a test male were introduced into courtship chambers. Courtship index was calculated as the percentage of time that males engaged in courtship activity for the first 10 min, and analyzed manually using LifeSong X. Copulation index was calculated as the percentage of flies that successfully copulated during the 30 min assay. For competitive courtship assays, GF and MB males were tracked with a small dot of paint on the thorax. A female was gently aspirated into a well-lit mating chamber with a pair of GF and MB males. The aggressive behavior was assayed as above described, and males that first copulated with the female was determined as the "winner".

**Drug treatment**. CDM treatment was carried out as previously depicted[32]. Briefly, 15 flies were grouped in vials and aged for 3–5 days before drug treatment. CDM (Sigma Aldrich, St. Louis, MO, USA) was dissolved in a 5% sucrose (wt/vol) solution, and the amount (200 µl) at the appropriate dilution was added to filter paper (Waterman, 3 mm) at the bottom of vials. Flies were then transferred into the vials for 24 h before the behavioral tests.

**HPLC/MS analysis**. The MS determination was performed on an Agilent MSD/QTOF 6545 system (Agilent Technologies, Germany) coupled with HPLC/1290II and the instrument parameters were set as follows: ESI source nebulizer gas ($N_2$) flow rate 6 L/min, temperature 300 ˚C, pressure 20 psig, sheath gas flow 10 L/min, temperature 320 ˚C, Vcap 3500 V, skimmer 65 V, OCT RF 750 V, fragmentor 145 V, nozzle 500 V, and mass scan on positive mode in range m/z 80–3200 while be calibrated by online standard reference ion 121.05 and 922.01. For chemical separation, water and acetonitrile of both consists 0.1% formic acid were used as mobile phase to wash sample out from a column (Agilent extend 300-C18, 4.6 ×150 mm, 3.5 µm) at flow rate 0.3 ml/min within 14 min as begin from acetonitrile 3% to 10 min 9%, 11 min 95% hold on 3 min then back to 3%.

**Quantitative real-time PCR**. RNA was isolated from 30 fly heads using TriZol reagent (Thermo Fisher). Ten micrograms of RNA were reversely transcribed using SuperScript IV First-Strand Synthesis System Kit (Thermo Fisher), according to the manufacturer's instructions. The primers used were previously described[65,66] and listed in Supplementary Table 1. Real-time PCR was performed using a LightCycler 96 (Roche). All values are the average of three independent replicates and normalized to actin mRNA.

**16 S rRNA sequencing and analysis**. Each sample with 1 g fly food or 20 guts was respectively collected, and sent to the Novogene Bioinformatics Technology Co., Ltd (Beijing, China). Total bacterial DNA extraction and sequencing was performed in accordance with standard protocols[30]. Briefly, DNA was amplified using the 515 f/806r primer set (515 f: 5′-GTGCCAGCMGCCGCGGTAA-3′, 806r: 5′-XXX GGACTACHVGGGTWTCTAAT-3′), which targets the V4 region of the bacterial 16 S rDNA. Pyrosequencing was conducted on an Illumina MiSeq. 2 × 250 platform according to published protocols[30]. Sample reads were assembled using mothur v1.32. Chimeric sequences were removed using the USEARCH software based on the UCHIME algorithm. The microbial diversity was analyzed using the QIIME software with Python scripts. Operational taxonomic units (OTUs) were picked using the de novo OTU picking protocol, with a 97% similarity threshold.

**Tissue dissection, staining, imaging, and quantification**. Brains of 4–6 days old adult males (Fig. 4e), and CNS of L2 or L3 larva (Fig. 5g–l) were dissected in Schneider's insect medium (S2) and fixed in 2% paraformaldehyde in S2 medium for 50–60 min at room temperature. After 4 × 10 min washing in PAT (0.5% Triton X-100, 0.5% bovine serum albumin in phosphate-buffered saline), tissues were blocked in 3% normal goat serum (NGS) for 90 min, then incubated in primary antibodies diluted in 3% NGS for 12–24 h at 4 ℃, then washed in PAT, and incubated in secondary antibodies diluted in 3% NGS for 1–2 days at 4 ℃. Tissues were then washed thoroughly in PAT and mounted for imaging. Antibodies used were rabbit anti-Tdc2 (1:400, pab0822-P, covalab, France), mouse anti-Bruchpilot (1:50, nc82, Developmental Studies Hybridoma Bank), and secondary Alexa Fluor 488 (1:500, A-11029, Invitrogen) and 568 antibodies (1:500, A-11004, Invitrogen). Samples were imaged at ×20 magnification on confocal microscopes (Zeiss 700 or 710) and processed with Fiji software. Monochrome images were rendered to emphasize differences in intensity. For each region of interest (ROI), a maximum z-projection of a fixed number of image stacks with Tdc2 signals was created, and the average fluorescence intensity was calculated for each sample using software ImageJ (1.52 v).

**Statistical analysis**. Statistical analysis is performed using GraphPad Prism 8 and indicated inside each figure legend. Layout of all figures used Adobe Illustrator CC 2019. Experimental flies and genetic controls were tested at the same condition, and data are collected from at least two independent experiments.
D'Agostino–Pearson normality test was used to verify the normal distribution of data. If normally distributed, two-tailed Student's t-test was used to compare two groups, and one-way ANOVA followed by Tukey's multiple comparisons test was used for comparisons of multiple groups. If not normally distributed, two-tailed Mann–Whitney U-test was performed to compare two groups of samples, while Kruskal–Wallis test followed by Dunn's multiple comparisons test was used for multiple comparisons among three or more groups.

**Reporting Summary**. Further information on research design is available in the Nature Research Reporting Summary linked to this article.

## Data availability
All data generated or analyzed during this study are included in the manuscript and its supplementary information files. All other relevant data supporting the findings of this study are available from the corresponding author upon reasonable request. Source data are provided with this paper.

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

## Acknowledgements
We thank the Bloomington Stock Center, Dr. Yufeng Wang and Dr. Yi Rao for fly stocks. This work was supported by grants from National Key R&D Program of China (2019YFA0802400 to Y.P.), the National Natural Science Foundation of China (31501175 to W.L., 31970943 and 31622028 to Y.P., and 81871121 to S.J.), and the Jiangsu Innovation and Entrepreneurship Team Program (to Y.P.).

## Author contributions
Y.J., S.J., K.H., and Y.Pang performed the behavioral experiments. L.G. and E.L. collected and analyzed the samples for HPLC experiments. Y.J. and C.H. collected samples for the real-time PCR experiment and analyzed the data. Y.J. and R.K. collected samples for 16 S rRNA sequencing and analyzed the data. Y.Pan, W.L., and E.T. conceived and supervised the project. Y.J., Y.Pan, and W.L. wrote the manuscript with input from all authors.

## Competing interests
The authors declare no competing interests.
