## [Peer Review File · Nature Communications]

Reviewers' comments:

Reviewer #1 (Remarks to the Author):

The manuscript by Jia and colleagues tackles a timely and important issue: how do gut bacteria act on the host to alter behavior. They choose to approach this complex problem using the powerful *Drosophila* system. Over the last years this system, which has been classically used to address molecular and developmental questions, has also been successfully used to tackle the impact of commensal bacteria on host physiology and behavior. It is nowadays a widely accepted system to identify mechanisms by which gut microbes act on the host.

The authors use this system to test if the fly microbiome affects aggression. They provide intriguing evidence that this could indeed be the case and attempt to provide a mechanistic basis for the observation.

Unfortunately, the manuscript is written in an unnecessary inflammatory way by casting doubts on the whole field of *Drosophila* microbiome and behavior with a special focus on a recent paper by Schretter et al.. This is especially problematic as the work of the authors is on itself not performed to the necessary high standards which would be required to sustain their claims. The manuscript is often contradictory and many arguments not really clear. Furthermore the authors lack the scholarly rigor to be able to critically discuss different aspects of work performed in the field which is directly relevant for their claims. They for example do not cite papers which have not been disputed. While the observation that gut microbes could affect aggression in *Drosophila* is interesting and relevant I can not support the manuscript in its current form.

Here are my major concerns:

- 1) The authors broadly attack the field of *Drosophila* microbiome and behavior without critically discussing it and providing sound evidence for their claim. While I agree that some observations reported in papers are based on weak effects and that there is currently some debate as the conditions in which specific behavioral effects can be observed, the broad claim that it is currently debatable if the *Drosophila* microbiome affects behavior is simply inflammatory and wrong. Especially as the authors do not cite key papers in the field and also later on, use the literature they attack to support their claims. Generally, I agree that the field is complex and that the exact conditions in which experiments are done heavily influence the outcome but the conclusion should be a call to more nuance and care in how the experiments are performed and not a broad casting in doubt of the literature. Especially when one owns experiments are all but careful and clear.
- 2) The generalized attack on the work of Schretter et al. is unnecessary, overblown and in my opinion unjustified. Especially, given that the authors themselves find that depending on how they analyze locomotion they can observe behavioral effects. Also to be able to say that effects in the literature and not reproducible one needs to do exactly the experiment performed by Schretter et al. Which the authors do not do.
- 3) The most likely explanation for the discrepancies in the literature is the fact that the impact of the microbiome on the host depends a lot on the diet. There is abundant data on this for development by the Leulier lab and in terms of behavior a key paper is Leitao-Goncalves et al. from the Ribeiro lab. The authors themselves highlight that by showing that the impact of the microbiome on development depends on the diet used (Suppl. Fig. 1). It is most likely that discrepancies in the literature can be explained by that. Also both locomotion and aggression are affected by nutritional state and this is mediated by octopamine signaling. The authors should therefore test different diets with different energy and protein content. They should also cite the papers linking microbiome to diet and behavior. Especially in *Drosophila*.

4) The experiments presented in the paper are generally badly controlled and described. For example, nowhere in the manuscript do the authors describe the bacterial load of the flies. It is therefore completely unclear if the flies they are testing have a microbiome, how many bacteria of which species they are carrying and to which extent the experimental treatments affect the microbes. For example, are the bacteria still in the adult flies given that they are provided to the eggs/larvae? Is the absence of effect on aggression due to providing the bacteria later after egg laying caused by adult flies having more or less bacteria? How efficient is the protocol to generate germ-free flies? The authors have to provide CFU's for the different bacteria in all the experiments which are being presented in the paper to make sure that the assumed bacterial load is indeed the reality. They then have to correlate the observed effects with the measured bacterial load.

5) I am very confused by the effect of the microbiome on the behavior of the flies. First, it is clear that different types of treatments affect octopamine. The most prominent being starvation. Which could account for many of the observed effects. The authors should therefore look at the transcriptional levels of genes responsible for octopamine synthesis in different diets.

6) Furthermore it is not clear to me how the authors explain that a small window in providing bacteria at a very early developmental stage can affect octopamine signaling. What is the level of octopamine and the transcription of Tdc2 in flies which have been provided with bacteria at later stages (with no effect)? How can they explain the developmental effect? Is it affecting neurons, gene transcription... I am aware that these are tough questions but the claim of the authors is quite exceptional and that required way more evidence than they provide.

7) I am not sure that the octopamine signaling manipulations show that the microbiome acts on aggression by changing octopamine signaling. It is currently totally possible that changing octopamine signaling affects aggression in parallel to the microbiome effect. The fact that octopamine is involved in modulating aggression in *Drosophila* has been shown a long time ago and this is therefore a good possibility. The authors should better design their experiments to exclude that possibility.

I also have some minor comments relates to the lack of explanations on how exactly aggression is being scored and analyzed but I will not go into those smaller comments as there are so many main comments which the authors should consider.

Overall the manuscript is making exaggerated claims about the lack of effects in the field of *Drosophila* behavior and the microbiome. It lacks appropriate support for that claim and most irritating is that the paper itself is contradictory in this claim. The authors do not cite key papers and do not discuss them in a scholarly fashion. But most critically their own conclusions are not properly supported and the experiments are very poorly controlled. The observation that aggression could be affected by the microbiome is interesting but needs more support. They also need to explore a possible link between diet and microbiome as a possible explanation of the effect they observe. Given this long list of shortcomings I can not support the publication of this work in *Nature Communications*.

Reviewer #2 (Remarks to the Author):

This manuscript by Jia et al. reports on the impact of the gut microbiome on *Drosophila* aggression and on the synthesis of octopamine. The topic is significance and results will be of interest to many in the *Drosophila* as well as gut-brain community. As several of the

experiments reach the opposite conclusion as studies already published, the authors take care to explain possible explanations for these differences. The addition of the live or head-killed bacteria is a useful control. Overall, the question is timely however details regarding the assays are not provided and therefore the conclusions may not be well founded. Inclusion of missing data or controls would provide clarification. Resolution of the following points is necessary to strengthen the manuscript.

Major points.

1. The foundation of the paper is the establishment of germfree flies. The authors need to show the results of their 16S rDNA PCR and nutrient agar plates for each genotype. The 16s rDNA PCR method is more sensitive and thus should be used for each line. Likewise, it isn't clear when the MB media is made with conventionally reared adult flies or *L. plantarius*, this needs to be specified and the re-colonization demonstrated by PCR. The effort it takes to generate four transgenic lines that are germfree and then cross these lines is well-appreciated. Nevertheless, it is critical to show in a supplemental figure that each line is GF.
2. What is the significance of repeated the yeast enrichment studies (SFig.1). Either provide a rationale for this study (did the authors ever test nutrition scarcity?) or leave it out.
3. Additional details on the aggression assay need to be provided as this is the main emphases of the paper. The methods describe "Latency of fighting was calculated by the time from the beginning of video recording to the first fight". What is "fight"? Do the authors mean lunge? Second, "lunging frequency was calculated as the number of lunging during the first 10 mins". Is this the first 10 mins after introduction into the chamber or the first 10 minutes after the first "fight"? The difference between these two approaches is large and if the first, could erase any differences between controls and GF flies.
4. Members of the *Drosophila* aggression community use different chambers and methods of interpreting the data. However, lunging frequency is not informative especially with the caveat above. The authors need to show the lunge number data even if it is a supplemental figure. Likewise, the fighting latency is meaningless unless "fighting" is defined.
5. The authors speculate less aggression is displayed in CR vs. GF pairs due to lower aggressive drive in GF males. It isn't clear what is meant by this statement. Do both males lunge in the CR-CR pairs? If one male displays the majority of lunges, one would expect this to carry over in the CR-GF pair. Other explanations include changes in pheromones in the GF males.
6. The rescue of aggression by re-colonization of bacteria is an interesting result and suggests the gut microbiome is required for nervous system development – even before the changes during pupariation – to generate wildtype levels of aggression.
7. As in Fig. 2, each assay in Fig. 3 must have the CR males tested for comparison. Likewise, Supplementary Fig. 5, must have the CR females for comparison and fighting frequency defined in females.
8. In Fig. 4, it isn't clear if the current results regarding *Tdc2* mRNA levels are different than the previous study as the CR controls are not provided in this study. CR mRNA levels must be provided for each gene tested. The MB flies are interesting but they are not controls. The authors realize this as on lines 195-196, it is written "the mRNA levels of *Tdc2* but not other

genes was significantly decreased in heads of GF females compared with that of CR females (Fig. 4b)” but then the correct controls are not shown.

Minor points.

1. Explain why *L. plantarum* was used.

2. line 179-180, the results in Fig. 3d indicate the differences between MB and GF males are statistically different which is more accurate than “prone”.

Reviewer #3 (Remarks to the Author):

In their manuscript, Jia, Hu, Jin, and colleagues report that the microbiota contributes to aggressive behavior in Canton-S *D. melanogaster* flies. By generating GF flies and comparing to conventionally reared flies, the authors show that while no global effects on locomotion and sleep, the microbiota contributes to aggressive behavior, especially in a competitive mating context. The authors go on to show that *Lactobacillus plantarum*, a member of the fly microbiota, is sufficient to rescue the phenotype in GF flies. Further, the authors show that, as previously reported, this behavior is driven by octopamine in flies’ brains, and that octopaminergic neurons are necessary and sufficient for the microbiota’s contribution to lunging behavior (aggression). This paper is well written and experiments are well designed, and results are exciting and of interest to the community. However, additional control experiments and analyses are required to substantiate several claims throughout in this paper.

Specific comments and questions–

- Line 64 – inconsistent results in studies are often the result of different microbiomes and/or experimental choices that can affect the microbiome. See (in mice)

<https://www.nature.com/articles/s41564-019-0407-8> for example. It is important to report the experimental methodology in full, including fly and bacterial strains, diet, rearing temperatures and humidity, time of day for behavioral assays, and any other experimental choices that may affect fly behavior.

- It is unclear what bacterial community was present in flies. Have the authors profiled the microbiome in these flies (CR vs MB)? What bacteria are present in “mixed bacteria”? It would be of importance, especially when comparing to Schretter et al 2018. Is *L. brevis* present in these flies?

- What concentration of bacteria (in CFU) were introduced to recolonize GF flies with mixed bacteria and *L. plantarum*? Methods are unclear.

- I suggest moving figure 1 to supplement as does not directly contribute to the conclusion presented in the title or the questions followed up on in the rest of the paper.

- Supplementary Figure 1 should specify what (CR, GF, MB) mean.

- What age of flies was used for locomotion experiments? What time of day were tests started? Was the time of tests the same for all groups (run side-by-side)?

- Light/dark status can affect phenotype. It would be best to present zeitgeber time rather than hours from start. In addition, data should be stratified by light and dark phase.

- Line 100 - It is great that the authors made sure to control for autoclaving media vs irradiation. Other experimental choices should be controlled for as well. E.g., Canton S vs

Oregon R, tegosept concentrations, different growth media (glucose vs molasses), and specific microbial populations or strains.

- Line 113 – As above, assuming that 0 is light and ~12 is the dark phase, it seems like the effects on locomotion are specific to whether it is the light or dark phase, and that they are relevant to changing cues. I am not sure it is fair to dismiss these as inconsistent and to conclude that there are no effects on locomotor behavior.

- It would be of interest to test locomotion in monocolonized/gnotobiotic flies, with *L. plantarum* and *L. brevis*.

- In aggression assays, were lunging events counted and presented for both flies in the arena or just one? Does the order of introduction to the arena matter? In GF vs CR (in methods says GF vs MB) assays, does one lung more than the other?

- Figure 2d – were these tested in arenas with conspecifics?

- Line 127 – by irradiated assay, I assume the authors mean irradiated media?

- Supplementary Figures 4 and 5 should be Supplementary Figures 5 and 4, respectively.

- Lines 143 and 150 – The figure reference should be Fig 2d and 3 rather than 1.

- It is unclear how or why the authors chose to test *Lactobacillus plantarum* for rescue. Where other bacteria tested? Was *Lactobacillus brevis*, previously shown to affect octopaminergic neurons, tested? What strain was used? Where was it isolated from? how were flies inoculated? When and how long?

- Line 163 – was courtship tested vs a conventionally reared virgin females or GF?

- Line 247 – the authors say that “inconsistent findings in Canton-S and Oregon-R flies demonstrated a trivial role, if any, of gut microbiome in modulating host locomotor behaviors”. In their paper, Schretter et al compared both CS and OR and show that the effect is indeed robust and holds in both lines. Is that the case for aggression? Does the microbiome contribute to aggression phenotype in OR flies in a similar manner to the results they report in CS flies?

- Line 262 – Schretter et al 2018 showed that *Lactobacillus brevis*, not *Lactobacillus plantarum* affects *Tdc2* expression (also, note there is a typo – not *L. plantarus*; same is specified in methods Line 312).

- In the reporting summary, the authors state for Replication that the results are representative of at least two independent experiments. However, it is not stated anywhere in the manuscript or methods. Was that indeed the case? In addition, I recommend making all data freely available on an Data Repository (e.g., Dryad, FigShare).

Our point-to-point responses are in blue.

Reviewer #1 (Remarks to the Author):

The manuscript by Jia and colleagues tackles a timely and important issue: how do gut bacteria act on the host to alter behavior. They choose to approach this complex problem using the powerful *Drosophila* system. Over the last years this system, which has been classically used to address molecular and developmental questions, has also been successfully used to tackle the impact of commensal bacteria on host physiology and behavior. It is nowadays a widely accepted system to identify mechanisms by which gut microbes act on the host.

The authors use this system to test if the fly microbiome affects aggression. They provide intriguing evidence that this could indeed be the case and attempt to provide a mechanistic basis for the observation. Unfortunately, the manuscript is written in an unnecessary inflammatory way by casting doubts on the whole field of *Drosophila* microbiome and behavior with a special focus on a recent paper by Schretter et al.. This is especially problematic as the work of the authors is on itself not performed to the necessary high standards which would be required to sustain their claims. The manuscript is often contradictory and many arguments not really clear. Furthermore the authors lack the scholarly rigor to be able to critically discuss different aspects of work performed in the field which is directly relevant for their claims. They for example do not cite papers which have not been disputed. While the observation that gut microbes could affect aggression in *Drosophila* is interesting and relevant I can not support the manuscript in its current form.

We thank the reviewer for acknowledgement of the significance of our findings and pointing out issues regarding to some experiments and also our writing. In this new version, we added substantial new experiments and controls, and carefully (almost completely) rewrote the manuscript. As there are too many changes, we did not provide the track-change version of the manuscript.

Here are my major concerns:

1) The authors broadly attack the field of *Drosophila* microbiome and behavior without critically discussing it and providing sound evidence for their claim. While I agree that some observations reported in papers are based on weak effects and that there is currently some debate as the conditions in which specific behavioral effects can be observed, the broad claim that it is currently debatable if the *Drosophila* microbiome affects behavior is simply inflammatory and wrong. Especially as the authors do not cite key papers in the field and also later on, use the literature they attack to support their claims. Generally, I agree that the field is complex and that the exact conditions in which experiments are done heavily influence the outcome but the conclusion should be a call to more nuance and care in how the experiments are performed and not a broad casting in doubt of the literature. Especially when one owns experiments are all but careful and clear.

Thanks for the comment and we now completely rewrite the manuscript, and focused on the relationship of gut microbiome and aggression, but not the discrepancy of our findings from Schretter et al.

2) The generalized attack on the work of Schretter et al. is unnecessary, overblown and in my opinion unjustified. Especially, given that the authors themselves find that depending on how they analyze locomotion they can observe behavioral effects. Also to be able to say that effects in the literature and not reproducible one needs to do exactly the experiment performed by Schretter et al. Which the authors do not do.

We completely rewrote the introduction and discussion. We added more experiments relating to locomotor behaviors and discussed the discrepancy between our results and findings by Schretter et al. We also pointed out an important factor to assay locomotor behaviors (below is a part of the discussion): we note that Schretter *et al.* tested walking speed of flies between ZT0 and ZT3; however, locomotor behaviors vary a lot during this period after morning peak (it peaks at ZT0, and decreases by ~80% at ZT3 [from ~200 mm/min to ~30 mm/min], see Fig. 2a, 2e and Supplementary Fig. 5c and 5f), which may contribute to their observed locomotor differences. For such reason, we suggest future test of locomotor behaviors in a shorter time window (*e.g.*, 30 min test between ZT1 and ZT2 with control and experimental flies tested simultaneously), or for a longer time (*e.g.*, average walking speed for 3 hours or even 24 hours as used in this study).

3) The most likely explanation for the discrepancies in the literature is the fact that the impact of the microbiome on the host depends a lot on the diet. There is abundant data on this for development by the Leulier lab and in terms of behavior a key paper is Leitao-Goncalves et al. from the Ribeiro lab. The authors themselves highlight that by showing that the impact of the microbiome on development depends on the diet used (Suppl. Fig. 1). It is most likely that discrepancies in the literature can be explained by that. Also both locomotion and aggression are affected by nutritional state and this is mediated by octopamine signaling. The authors should therefore test different diets with different energy and protein content. They should also cite the papers linking microbiome to diet and behavior. Especially in *Drosophila*.

Thanks for this wonderful suggestion, and we now performed more experiments on how diet is involved in the microbiome-mediated changes of development and aggression (new Figure. 5). We tested how different concentrations of yeast in food could affect development, expression of *Tdc2* and intermale aggression; however, we were not able to test the effects of other diets (*e.g.*, different concentrations of sugar) on aggression due to the limited scope of this study. We also cited related papers on how microbiome could affect development, metabolism and behavior in response to nutrition environment, and discussed how microbiome and diet could jointly modulate aggression in flies.

4) The experiments presented in the paper are generally badly controlled and described. For example, nowhere in the manuscript do the authors describe the bacterial load of the flies. It is therefore completely unclear if the flies they are testing have a microbiome, how many bacteria of which species they are carrying and to which extent the experimental treatments affect the microbes. For example, are the bacteria still in the adult flies given that they are provided to the eggs/larvae? Is the absence of effect on aggression due to providing the bacteria later after egg laying caused by adult flies having more or less bacteria? How efficient is the protocol to generate germ-free flies? The authors have to

provide CFU's for the different bacteria in all the experiments which are being presented in the paper to make sure that the assumed bacterial load is indeed the reality. They then have to correlate the observed effects with the measured bacterial load.

We totally agree with the reviewer and should put these control data into the manuscript at the first place. We have extensive expertise to generate GF and gnotobiotic flies as shown in our recent papers (Liu et al., Scientific Reports and Su et al., BMC Microbiology). Axenia in fly intestines was confirmed by plating the homogenates on nutrient agar plates or performing 16S rDNA PCR (Fig. 1b and Supplementary Fig. 1). Moreover, we performed 16S ribosomal DNA (rDNA) sequencing to analyze the bacterial community structure of *Drosophila* and diet microbiota. Consistent with previous studies, our laboratory-reared flies were typically associated with 5 to 30 species that primarily consist of the *Acetobacter* and *Lactobacillus* genera (Supplementary Fig. 3a-d).

5) I am very confused by the effect of the microbiome on the behavior of the flies. First, it is clear that different types of treatments affect octopamine. The most prominent being starvation. Which could account for many of the observed effects. The authors should therefore look at the transcriptional levels of genes responsible for octopamine synthesis in different diets.

We tested the relative mRNA level of *Tdc2* in CR, GF and MB males raised in diets with 0.5%, 2.5% or 10% yeast. We found that the levels of *Tdc2* mRNA were significantly elevated in CR and MB adult brains if raised throughout development with higher yeast concentration, while levels of *Tdc2* mRNA in GF males were comparable under feeding with different yeast concentrations (0.5%, 2.5% and 10%), and lower than those in CR and MB males raised on food with 2.5% or 10% yeast (Fig. 5d). These new results demonstrated that the microbiome promotes male aggression in a diet-dependent manner, and also revealed an important role of diet during development for establishing a regular level of aggression in adult males.

6) Furthermore, it is not clear to me how the authors explain that a small window in providing bacteria at a very early developmental stage can affect octopamine signaling. What is the level of octopamine and the transcription of *Tdc2* in flies which have been provided with bacteria at later stages (with no effect)? How can they explain the developmental effect? Is it affecting neurons, gene transcription... I am aware that these are tough questions but the claim of the authors is quite exceptional and that required way more evidence than they provide.

Emerging studies showed that early-life exposure to microbiome promotes the maturation and homeostasis of the organism systems through the release of microbial products, imparting long-lasting effects on host physiology and behaviors. We added a paragraph in discussion on how diet and microbiome during development could affect adult behavior and cited key papers. We also showed that the level of *Tdc2* mRNA in adults that were inoculated with bacteria 96 hours post oviposition was significantly lower compared to earlier-life colonization (Fig. 5f), consistent with the aggression levels these males displayed.

7) I am not sure that the octopamine signaling manipulations show that the microbiome acts on aggression by changing octopamine signaling. It is currently totally possible that changing octopamine signaling affects aggression in parallel to the microbiome effect. The fact that octopamine is involved in modulating aggression in *Drosophila* has been shown a long time ago and this is therefore a good possibility. The authors should better design their experiments to exclude that possibility.

We are aware that manipulating *Tdc2* neuronal activity in GF and MB males does not sufficiently to conclude if microbiome modulates aggression by changing octopamine signaling, and we did not make that claim. We presented these data in the background of other results shown in Fig. 4 that tested OA signaling using multiple methods.

I also have some minor comments relates to the lack of explanations on how exactly aggression is being scored and analyzed but I will not go into those smaller comments as there are so many main comments which the authors should consider.

We carefully examined all method parts and provided necessary details.

Overall the manuscript is making exaggerated claims about the lack of effects in the field of *Drosophila* behavior and the microbiome. It lacks appropriate support for that claim and most irritating is that the paper itself is contradictory in this claim. The authors do not cite key papers and do not discuss them in a scholarly fashion. But most critically their own conclusions are not properly supported and the experiments are very poorly controlled. The observation that aggression could be affected by the microbiome is interesting but needs more support. They also need to explore a possible link between diet and microbiome as a possible explanation of the effect they observe. Given this long list of shortcomings I can not support the publication of this work in *Nature Communications*.

We hope this revision with substantial new experiments and careful rewriting has improved the manuscript and addressed most if not all concerns.

Reviewer #2 (Remarks to the Author):

This manuscript by Jia et al. reports on the impact of the gut microbiome on *Drosophila* aggression and on the synthesis of octopamine. The topic is significance and results will be of interest to many in the *Drosophila* as well as gut-brain community. As several of the experiments reach the opposite conclusion as studies already published, the authors take care to explain possible explanations for these differences. The addition of the live or head-killed bacteria is a useful control. Overall, the question is timely however details regarding the assays are not provided and therefore the conclusions may not be well founded. Inclusion of missing data or controls would provide clarification. Resolution of the following points is necessary to strengthen the manuscript.

We thank the reviewer for being very positive on our manuscript. In this new version, we added substantial new experiments and controls, and carefully (almost completely) rewrote the manuscript. As there are too many changes, we did not provide the track-change version of the manuscript.

Major points.

1. The foundation of the paper is the establishment of germfree flies. The authors need to show the results of their 16S rDNA PCR and nutrient agar plates for each genotype. The 16s rDNA PCR method is more sensitive and thus should be used for each line. Likewise, it isn't clear when the MB media is made with conventionally reared adult flies or *L. plantarius*???, this needs to be specified and the re-colonization demonstrated by PCR. The effort it takes to generate four transgenic lines that are germfree and then cross these lines is well-appreciated. Nevertheless, it is critical to show in a supplemental figure that each line is GF.

We have now provided these control experiments. Axenia in fly intestines was confirmed by plating the homogenates on nutrient agar plates or performing 16S rDNA PCR (Fig. 1b and Supplementary Fig. 1). Moreover, 16S ribosomal DNA (rDNA) sequencing revealed the bacterial community structure of *Drosophila* and diet microbiota (Supplementary Fig. 3). Consistent with previous studies, our laboratory-reared flies were typically associated with 5 to 30 species that primarily consist of the *Acetobacter* and *Lactobacillus* genera.

2. What is the significance of repeated the yeast enrichment studies (SFig.1). Either provide a rationale for this study (did the authors ever test nutrition scarcity?) or leave it out.

Thanks for this suggestion. We have now rewritten this part and added additional results on how diet and microbiome during development could jointly modulate adult aggression. The new added results are now put into Figure 5 and also being discussed.

3. Additional details on the aggression assay need to be provided as this is the main emphases of the paper. The methods describe "Latency of fighting was calculated by the time from the beginning of video recording to the first fight". What is "fight"? Do the authors mean lunge? Second, "lunging frequency was calculated as the number of lunging during the first 10 mins". Is this the first 10 mins after introduction into the chamber or the first 10 minutes after the first "fight"? The difference between these two approaches is large and if the first, could erase any differences between controls and GF flies.

We are sorry for not adding sufficient information in the method parts, and now carefully examined all method parts and provided necessary details. The latency of fighting was calculated by the time from the beginning of video recording to the first lunging, while the lunging frequency was now calculated as the number of lunging per minute after lunging initiation. Lunging frequency was analyzed for a total of 10 minutes after lunging initiation.

4. Members of the *Drosophila* aggression community use different chambers and methods of interpreting the data. However, lunging frequency is not informative especially with the caveat above. The authors need to show the lunge number data? even if it is a supplemental

figure. Likewise, the fighting latency is meaningless unless “fighting” is defined.

We have added necessary details in the methods part as mentioned above, and we put all data into a source data file as a supplemental file now.

5. The authors speculate less aggression is displayed in CR vs. GF pairs due to lower aggressive drive in GF males. It isn't clear what is meant by this statement. Do both males lunge in the CR-CR pairs? If one male displays the majority of lunges, one would expect this to carry over in the CR-GF pair. Other explanations include changes in pheromones in the GF males.

Thanks for this comment, and we have rewritten this part as following: We also found that GF-CR pairs showed decreased lunging frequency compared to CR-CR pairs and increased fighting latency (Fig. 1d and e, and Movie S3), which could be due to a lower aggressive drive in the GF male; alternatively, there may be pheromone changes from the GF male's cuticle that decreased the CR opponent's aggressive behaviors (Hefetz et al., 2011).

6. The rescue of aggression by re-colonization of bacteria is an interesting result and suggests the gut microbiome is required for nervous system development – even before the changes during pupariation – to generate wildtype levels of aggression.

Thanks for your acknowledgement in this result, and actually we provided more evidence on how microbiome and diet during development jointly modulate adult aggression (Figure 5), and also added discussion about it.

7. As in Fig. 2, each assay in Fig. 3 must have the CR males tested for comparison. Likewise, Supplementary Fig. 5, must have the CR females for comparison and fighting frequency defined in females.

For certain experiment, like a competitive courtship assay, we used GF and MB as they are probably better comparisons, as MB flies went through the same sterilization process as the GF flies. In this revision, we used CR, GF and MB (and *L.p* recolonized flies in some cases) in most of our results now.

8. In Fig. 4, it isn't clear if the current results regarding Tdc2 mRNA levels are different than the previous study as the CR controls are not provided in this study. CR mRNA levels must be provided for each gene tested. The MB flies are interesting but they are not controls. The authors realize this as on lines 195-196, it is written “the mRNA levels of Tdc2 but not other genes was significantly decreased in heads of GF females compared with that of CR females (Fig. 4b)” but then the correct controls are not shown.

Please referred to our last response above. CR female was a typo in the previous manuscript, and we corrected it now. We now provided relative expression of Tdc2 mRNA of CR, GF and MB flies under different conditions (new Figure 5d).

Minor points.

1. Explain why *L. plantarum* was used.

L. plantarum is more common and widely used in previous studies, and rich in the fly gut (Supplementary Figure 3 and source data file). Previous studies also showed that *L.p* is sufficient to recapitulate the natural microbiome growth-promoting effect (Storelli et al., 2011, Cell Metablism).

2. line 179-180, the results in Fig. 3d indicate the differences between MB and GF males are statistically different which is more accurate than “prone”.

We changed the sentence to “MB males had significantly more chances to gain copulation with females compared to GF males”

Reviewer #3 (Remarks to the Author):

In their manuscript, Jia, Hu, Jin, and colleagues report that the microbiota contributes to aggressive behavior in Canton-S *D. melanogaster* flies. By generating GF flies and comparing to conventionally reared flies, the authors show that while no global effects on locomotion and sleep, the microbiota contributes to aggressive behavior, especially in a competitive mating context. The authors go on to show that *Lactobacillus plantarum*, a member of the fly microbiota, is sufficient to rescue the phenotype in GF flies. Further, the authors show that, as previously reported, this behavior is driven by octopamine in flies’ brains, and that octopaminergic neurons are necessary and sufficient for the microbiota’s contribution to lunging behavior (aggression). This paper is well written and experiments are well designed, and results are exciting and of interest to the community. However, additional control experiments and analyses are required to substantiate several claims throughout in this paper.

We thank the reviewer for being very positive on our manuscript. In this new version, we added substantial new experiments and controls, and carefully (almost completely) rewrote the manuscript. As there are too many changes, we did not provide the track-change version of the manuscript.

Specific comments and questions–

- Line 64 – inconsistent results in studies are often the result of different microbiomes and/or experimental choices that can affect the microbiome. See (in mice) <https://www.nature.com/articles/s41564-019-0407-8> for example. It is important to report the experimental methodology in full, including fly and bacterial strains, diet, rearing temperatures and humidity, time of day for behavioral assays, and any other experimental choices that may affect fly behavior.

We have carefully examined the method part and added all necessary experimental details into the method part. In addition, we revealed in this revision that diet is also very important for assaying the role of microbiome in fly behaviors. We also discussed the discrepancy of

locomotor data from ours and Schretter et al.

- It is unclear what bacterial community was present in flies. Have the authors profiled the microbiome in these flies (CR vs MB)? What bacteria are present in “mixed bacteria”? It would be of importance, especially when comparing to Schretter et al 2018. Is *L. brevis* present in these flies?

We thank the review for pointing out these very important experiments. Using 16S rDNA high-throughput sequencing, we examined the profile of the microbiome in CR flies (mixed bacteria) and fly food. We found that our laboratory-reared flies were typically associated with up to 30 species that primarily consist of the Acetobacteraceae and Lactobacillaceae family (Supplementary Fig. 3a and b). As we added recolonized GF flies with the above mixed bacteria to generate MB flies, we assume that MB flies also have these mixed bacteria as we revealed. We found that *L. brevis* was present in our flies, despite of lower level of abundance compared with *L. plantarum* (Supplementary Figure 3 and source data file).

- What concentration of bacteria (in CFU) were introduced to recolonize GF flies with mixed bacteria and *L. plantarum*? Methods are unclear.

We used 0.1 OD₅₉₅ bacterial cells (~ 10⁷ CFU) to recolonize GF flies. All these details were added into the methods part.

- I suggest moving figure 1 to supplement as does not directly contribute to the conclusion presented in the title or the questions followed up on in the rest of the paper.

We thank the review for this suggestion. We now reorganized figures to make the manuscript more focused on aggression, but we believe that it is necessary to show our locomotor behavioral data (now as Fig. 2), which reveals very specific effect of microbiome on aggressive behaviors.

- Supplementary Figure 1 should specify what (CR, GF, MB) mean.

We now added all necessary descriptions into figure legends.

- What age of flies was used for locomotion experiments? What time of day were tests started? Was the time of tests the same for all groups (run side-by-side)?

We used 4-6 days old group-housed flies (males or females) for locomotor tests. As we test locomotor behavior for 24 hours, we started video recording starting from ZT0 (9am in our lab). We have 6 cameras and always record the experimental and control groups simultaneously.

- Light/dark status can affect phenotype. It would be best to present zeitgeber time rather than hours from start. In addition, data should be stratified by light and dark phase.

We appreciate this suggestion, and now changed our sleep data using the zeitgeber time. However, the 24-hr locomotor data (walking speed) was assayed under constant light condition for easy video tracking and analysis, thus could not use the zeitgeber time. Note that flies still showed circadian rhythm during this 24-hr test as they are previously entrained under 12h:12h conditions.

- Line 100 - It is great that the authors made sure to control for autoclaving media vs irradiation. Other experimental choices should be controlled for as well. E.g., Canton S vs Oregon R, tegosept concentrations, different growth media (glucose vs molasses), and specific microbial populations or strains.

We tried to use more control experiments especially given that our locomotion data are not consistent with Schretter *et al.* We now used three different protocols to generate GF flies, and we compared aggression in wild type Canton-S and Oregon-R. We also compared different diets but focus on yeast as we found a crucial role of yeast feeding during a critical developmental period for modulating aggressive behaviors.

- Line 113 – As above, assuming that 0 is light and ~12 is the dark phase, it seems like the effects on locomotion are specific to whether it is the light or dark phase, and that they are relevant to changing cues. I am not sure it is fair to dismiss these as inconsistent and to conclude that there are no effects on locomotor behavior.

The 24-hr walking speed data is acquired under constant light condition (for ease of video recording), although flies still showed locomotor rhythms. We did notice that locomotor differences (if counted every 10 min) were mostly found during the morning and evening peaks, which are not consistently higher or lower, thus the average walking speed in 24 hr was not significantly different.

We note that Schretter *et al.* tested walking speed of flies between ZT0 and ZT3; however, locomotor behaviors vary a lot during this period after morning peak (it peaks at ZT0, and decreases by ~80% at ZT3 [from ~200 mm/min to ~30 mm/min] in our study, see Fig. 2a, 2e and Supplementary Fig. 5c and 5f), which may contribute to their observed locomotor differences.

We have discussed this and proposed future test of locomotor behaviors in a shorter time window (*e.g.*, 30 min test between ZT1 and ZT2 with control and experimental flies tested simultaneously), or for a longer time (*e.g.*, average walking speed for 3 hours or even 24 hours as used in this study).

- It would be of interest to test locomotion in monocolonized/gnotobiotic flies, with *L. plantarum* and *L. brevis*.

We have tested the locomotion of *L.p.*-associated flies, and found that sleep amounts and walking speeds in CR, GF, MB, *L.p.* flies are not significantly different from each other.

- In aggression assays, were lunging events counted and presented for both flies in the arena or just one? Does the order of introduction to the arena matter? In GF vs CR (in methods says GF vs MB) assays, does one lung more than the other?

Lunging events were counted and presented for two flies in the arena. Flies were introduced to the arena sequentially within a minute, but we did not know if the order of introduction matters as we did not distinguish lungs by the two flies due to time-consuming manual analysis of aggression.

- Figure 2d – were these tested in arenas with conspecifics?

Yes, these were tested in arenas between two males from the same treatment (e.g., aggression between two CR males, now moved to Fig. 1f).

- Line 127 – by irradiated assay, I assume the authors mean irradiated media?

Yes, it is with irradiated medium, and we described the full protocol to generate GF flies in Supplementary Figure 2.

- Supplementary Figures 4 and 5 should be Supplementary Figures 5 and 4, respectively.

We have re-organized figures and carefully examined their order throughout.

- Lines 143 and 150 – The figure reference should be Fig 2d and 3 rather than 1.

Sorry for the typos, and we have carefully examined every parts of the manuscript now.

- It is unclear how or why the authors chose to test *Lactobacillus plantarum* for rescue. Where other bacteria tested? Was *Lactobacillus brevis*, previously shown to affect octopaminergic neurons, tested? What strain was used? Where was it isolated from? how were flies inoculated? When and how long?

Laboratory-reared flies were typically associated with approximately ~30 bacterial species that primarily consist of the *Acetobacter* and *Lactobacillus* genera. *L. plantarum* colonize in a range of hosts, including mammals and human. It is more common and widely used in previous studies, and rich in the fly gut (Supplementary Figure 3 and source data file). Previous studies also showed that *L.p* is sufficient to recapitulate the natural microbiome growth-promoting effect (Storelli et al., 2011, Cell Metablism). *L.p* was isolated from the gut of *Drosophila* with the Genbank accession number: KY038178. We add the bacterial pellet into the fly food after chemical sterilization of fly embryos. We now added all necessary details into the methods part.

- Line 163 – was courtship tested vs a conventionally reared virgin females or GF?

We used CR females as courtship target as there might be pheromone changes in GF females.

- Line 247 – the authors say that “inconsistent findings in Canton-S and Oregon-R flies demonstrated a trivial role, if any, of gut microbiome in modulating host locomotor

behaviors”. In their paper, Schretter et al compared both CS and OR and show that the effect is indeed robust and holds in both lines. Is that the case for aggression? Does the microbiome contribute to aggression phenotype in OR flies in a similar manner to the results they report in CS flies?

We have discussed the discrepancy of locomotor data from ours and Schretter et al. OR males in our lab rarely fight against each other (Fig. S6), and due to such low level of aggression in OR males, we could only make arguments on microbiome’s role in aggression based on CS flies. Intriguingly, even some strains of CS flies display a weak fighting, and we used the CS strain from Dr. Yi Rao’s lab in Peking University (Zhou et al., 2008, Nature Neuroscience) throughout this study.

-Line 262 – Schretter et al 2018 showed that *Lactobacillus brevis*, not *Lactobacillus plantarum* affects Tdc2 expression (also, note there is a typo – not *L. plantarus*; same is specified in methods Line 312).

Sorry for the typos, and we now carefully examined the manuscript throughout. We used *L.p* in this study for reasons stated above.

-In the reporting summary, the authors state for Replication that the results are representative of at least two independent experiments. However, it is not stated anywhere in the manuscript or methods. Was that indeed the case?

Yes, all tests are representative of at least two independent experiments. We stated this in the methods/statistics part. We have now carefully examined the method part and added as many experimental details as we could.

In addition, I recommend making all data freely available on a Data Repository (e.g., Dryad, Fig Share).

We now provided all data into the source data file and also provided representative videos.

Reviewers' comments:

Reviewer #1 (Remarks to the Author):

Jia and colleagues have substantially improved the manuscript. The current version makes a more convincing case for the microbiome affecting aggression in *Drosophila* and is also much more balanced in tone compared to the previous manuscript. I personally find it very exciting that the authors can show that there is a diet-microbiome interaction in the juvenile animal which leads to alterations in adult behavior. This is an important advance as the authors also provide evidence that the imprinting happens at the level of octopamine signaling. This opens up a path towards mechanistically dissecting how diet microbiome interactions during development impinge on adult behavior.

There are however three technical issues which in my opinion should be addressed before I can support the publication of the manuscript. I also think that a set of straight forward experiments could lead to a much better understanding of how the microbiome leads to alterations in octopamine signaling and what is the nature of these changes.

For the technical issues:

1) One of the well-documented effects of the microbiome on *Drosophila* physiology is its interaction with the *Wolbachia* endosymbiont (see for example Hughes et al., PNAS 2014, Ye et al. in Journal of Invertebrate Pathology 2017, Fromont et al., Molecular Ecology 2019). Given that the authors find that the flies they are using harbor *Wolbachia* and *Wolbachia* has been shown to alter aggression by altering octopamine signaling in flies (Rohrscheib et al. Appl. Environmental Microbiology 2015 – cited by the authors) there is a strong possibility that the effect of the microbiome on aggression might be mediated by *Wolbachia*. In such a scenario the microbiome manipulations would alter the *Wolbachia* titers which would then lead to the previously documented changes in octopamine signaling and aggression. The authors should show that the phenomenon they describe is not related to the mechanism described by Rohrscheib and colleagues. This is easily done by showing that the effect of the microbiome on octopamine signaling and aggression is present in flies which are free of *Wolbachia* or other endosymbionts.

2) At this stage, I still have some concerns regarding the specificity of the effect observed by the authors. Both at the level of behavior as well as mechanism. Therefore in my opinion it is important to look into the following points:

a) Is the change in aggression levels due to the flies now performing other behaviors instead of aggressive ones? I am mainly concerned about feeding behavior competing with the aggressive drive of the flies. We have shown that the microbiome and dietary yeast alter feeding behavior in flies. Could it be that the flies which show low aggression levels are spending most of their time feeding or performing other behaviors like grooming? Then the effect of the microbiome would not be on aggression directly but rather on other behaviors which would then compete with the display of aggressive behaviors. The authors should therefore characterize the behavioral effect of microbiome manipulations in the flies in which they score aggression to test if the effect is mainly on aggression.

b) I am puzzled that changing octopamine biosynthesis alters aggression specifically. Octopamine has been shown to affect all kinds of behaviors. An open question to me is: Is the effect of the microbiome only affecting a subset of neurons altering aggression or general octopaminergic control of behavior like locomotion, flight, egg-laying, etc.? Maybe that is where the authors are going with the analysis of locomotion but they do not necessarily spell out clearly if their findings suggest that the microbiome manipulation only affects a subset of behaviors controlled by octopamine or all behaviors controlled by octopamine. If it is a specific effect on aggression how do the authors explain it? Does it mean that the effect of the microbiome during early development only affects specific

neurons? If yes which and how? This can be tested experimentally and would be an important addition to the paper. This would require a more thorough analysis of the anatomical impact of microbiome manipulations than currently done in Figure 4 (it is difficult to see anything in the current brain stainings).

c) The authors suggest that the main effect of the microbiome is to alter Tdc2 levels. If they could show that they can rescue the effect of removing the microbiome on aggression, by overexpressing Tdc2 acutely in the adult from a transgene in GF flies, that would show that no other developmental effect like changes in circuit architecture is at play.

3) I am puzzled that the authors find no effects of adding heat-killed bacteria or the supernatant of the bacterial culture to the flies. Especially given that a possible explanation of the effect of the microbiome could be the increase in protein absorption in development. Obviously, heat-killed bacteria would also have proteins. These are not straight forward experiments as one has to make sure that one adds the same level of bacterial material as present in the fly at the stage when it is provided. One has to take into account that one provides much higher titers than when inoculating the flies with living bacteria as heat-killed bacteria do not grow exponentially. Did the authors do this? I could not find any description of how that experiment was done in the manuscript. I think it is missing in the materials and methods. The authors should therefore describe the experiment in the methods section and ensure that the level of heat-killed bacteria material is the same as present in the food when flies are alive. This very likely means providing heat-killed material repeatedly throughout development and at high levels. This has been done by us or the Leulier lab in our most recent publications.

Conceptually:

In my opinion, the finding that the effect of the microbiome on adult behavior results from a dietary interaction with the microbiome at an early developmental stage is exciting and intriguing. There is a very straight forward hypothesis which emerges from work in the field, including ours, and which the authors could easily test and would point to a clear mechanism by which the microbiome alters behavior. The most straight forward hypothesis is that the microbiome (specifically Lp) increases the uptake of amino acids from the diet which then alters octopamine biosynthesis in the adult. The authors can easily test this hypothesis by expanding the experiments presented in Figure 5e. By switching flies to diets with different levels of yeast or amino acids at the different time points after egg laying (same as the ones when they provide the microbes) they should be able to test if they can i) revert the effect of the microbiome and ii) mimics the effect of the microbiome. An increase beyond 10% of yeast during the 0-48h AEL time window should lead to changes in Tdc2 expression and aggression in GF animals. Conversely, a reduction to 0.5% or 2.5% of yeast when the animals are associated with the microbiome should abolish the effect of the microbiome on aggression. This could show that the effect of the microbiome is specifically to increase AA absorption in the juvenile leading to a change in neuromodulation.

Also, the authors should test if a continuous association with the microbiome is required to give an effect on aggression. What happens if the flies are manipulated to lose their microbiome at 24h, 36h, and 48h after being associated with microbes at 36h AEL.

These straight forward experiments should be able to clarify if the microbiome mainly acts on adult aggression by altering AA uptake during development and what is the exact time window when the microbiome acts.

Finally, I would like to close by encouraging the authors to expand their introduction and discussion to include aspects which I think expand the impact of their paper by making it more interesting to a wider audience. There is a growing recognition that the effect of the microbiome on brain function and behavior can be best understood in the context of the

interaction of the microbiome with diet. The current work makes this case very nicely too and I think that the authors could profit from placing their paper in this context more explicitly. This is a core finding and makes total sense in the context of how many people think about the problem in the field. For relevant literature, they can look at work cited in reviews of John Cryan (e.g. Sandhu et al. *Transl. Research* 2017) and our recent review (Ezra-Nevo et al. *Current Opinion in Neurobiology* 2020).

I also think that there are two important big questions which the authors hardly discuss but which are very pertinent given their findings. First, it would be interesting if the authors could discuss how the microbiome could act on diet to change octopamine signaling, how these changes could be propagated to the adult stage, and most importantly why this is the case. Are the authors looking at an adaptive mechanism or is this a pathological situation? If I might venture to propose an idea: One could envisage that animals exposed to a rich diet are more likely to also be better mates than animals exposed to malnutrition. Therefore it would make sense to have them be more aggressive and hence dominant and more likely to pass on their offspring. But here we are in the realm of speculation. This is just an idea for the authors to interpret their data, which I do not expect them at all to pick up.

To finalize I think this is an exciting manuscript with interesting discoveries. I congratulate the authors for their work in such difficult times. Some technical issues remain which could severely alter the validity and interpretation of the data but they should be straight forward to address.

Carlos Ribeiro

Reviewer #2 (Remarks to the Author):

This is a revised manuscript by Jia et al., describing the impact of the gut microbiome on *Drosophila* aggression and on the production of octopamine. The topic remains significant and the results will be of interest to the gut-brain community as a whole. The language and writing of the paper as a whole has improved with more information on rationale and less comparisons with experiments from other groups.

Many of the major and minor points were addressed in this revision. Several remain which need to be need to be attended to before I can recommend publication.

Major points.

1. To demonstrate that germfree flies were generated, the authors now show 16S rDNA PCR results and growth on nutrient agar plates. The nutrient agar plates in Fig. 1B are too small and are not labeled. This can easily be fixed or enlarged and moved to a supplemental figure. 16S rDNA PCR results are shown for the Canton S wildtype and GF lines – but not for Tdc2-Gal4, UAS-TrpA1, or UAS-NaChBac. In my previous review, I wrote to the authors “it is critical to show in a supplemental figure that each line is GF”. The PCR experiments were not provided to demonstrate that the Gal4 line and UAS lines or alternatively the progeny were GF and they need to be, this is a fundamental part of science.

2. Additional information on the aggression assay was provided but this information is not satisfactory. The authors write that the lunge number was manually counted for 30 minutes. I wrote in my previous review “The authors need to show the lunge number data even if it is a supplemental figure.” This was not done. In the rebuttal letter, the authors state, “we put all

data into a source data files as a supplemental file now. I have the supplemental excel files (which is easy to follow and clear) in front of me. The 30 min lunge number is not provided, still only 10 minutes.

The authors ran the assay for 30 mins, quantified all lunges within 30 mins as the materials and methods say, and I simply requested that the lunge number for all 30 mins be placed in a supplemental figure. This does not require any new experiments, gives an accurate picture of the fighting behavior during the fight assay which is important due to previous reports and could highlight new interesting findings.

Related: please add the median line to scatterplots on all figures.

3. The RO1 interest data does not make sense in its current form. The authors need to put the RO1 box around smaller areas. For RO1-1, draw a box around a pair of neurons on one side, a single neuron, or a very small cluster of neurons and redo the quantification. It is apparent there are neurons at the top of the box that are expressing Tdc2 weakly. If the authors are trying to emphasize this area, then draw a box there but not what is currently used. The same should be applied to ROI-2.

4. The Tdc2-Kir2.1 experiments were previously done by Hoyer 2008 in CR control males. Therefore, this experiment is not supportive in this situation. If MB male aggression was different than CR males this experiment would be useful. This data could be removed or if kept, then make sure to describe in the text that this experiment and reference the Hoyer paper.

Minor:

Line 112, lunges not “lungs”

Reviewer #4 (Remarks to the Author):

I am assigned as an additional reviewer to evaluate the revised manuscript, especially based on whether the reviewer 3's concerns were addressed appropriately.

In this revised manuscript, the authors seemed to perform many experiments to support their claim and revise the text accordingly. In my point of view, the authors successfully addressed many concerns raised during the initial review. 16S data was nicely added to discuss the microbiome of their fly strains. Dietary manipulations increased the generality and robustness of the idea behind the early-life environment vs adult behaviour. Methods and legends are now firmly described.

The points which was not addressed in the revised manuscript;

1, To discuss the discrepancy, it would be better to use Oregon R to see locomotion. This is pointed out by R3 and other reviewers but still not included.

2, Using *L. plantarum* as a “representative gut microbiota” is not well justified. They mentioned their flies contain Lactobacillaceae as a dominant family, and that *L. plantarum* is a dominant bacterial species. This seems not the case as the majority of gut microbiome is from Acetobacteraceae (Fig. S3). I would like to recommend Acetobacter/Commensalibacter isolated from their flies to test the behaviour. Otherwise, *L. brevis* should have been used as pointed by R3. Although this is not mandatory, it is slightly disappointing to lose the opportunity to discuss species specificity/generality, and to discuss the above-mentioned discrepancy.

Our point-to-point responses are in blue.

Reviewer #1 (Remarks to the Author)

Jia and colleagues have substantially improved the manuscript. The current version makes a more convincing case for the microbiome affecting aggression in *Drosophila* and is also much more balanced in tone compared to the previous manuscript. I personally find it very exciting that the authors can show that there is a diet-microbiome interaction in the juvenile animal which leads to alterations in adult behavior. This is an important advance as the authors also provide evidence that the imprinting happens at the level of octopamine signaling. This opens up a path towards mechanistically dissecting how diet microbiome interactions during development impinge on adult behavior.

There are however three technical issues which in my opinion should be addressed before I can support the publication of the manuscript. I also think that a set of straight forward experiments could lead to a much better understanding of how the microbiome leads to alterations in octopamine signaling and what is the nature of these changes.

For the technical issues:

1) One of the well-documented effects of the microbiome on *Drosophila* physiology is its interaction with the *Wolbachia* endosymbiont (see for example Hughes et al., PNAS 2014 <https://www.pnas.org/content/111/34/12498>, Ye et al. in *Journal of Invertebrate Pathology* 2017, Fromont et al., *Molecular Ecology* 2019). Given that the authors find that the flies they are using harbor *Wolbachia* and *Wolbachia* has been shown to alter aggression by altering octopamine signaling in flies (Rohrscheib et al. *Appl. Environmental Microbiology* 2015 – cited by the authors) there is a strong possibility that the effect of the microbiome on aggression might be mediated by *Wolbachia*. In such a scenario the microbiome manipulations would alter the *Wolbachia* titers which would then lead to the previously documented changes in octopamine signaling and aggression. The authors should show that the phenomenon they describe is not related to the mechanism described by Rohrscheib and colleagues. This is easily done by showing that the effect of the microbiome on octopamine signaling and aggression is present in flies which are free of *Wolbachia* or other endosymbionts.

We thank the reviewer for this valuable suggestion. To rule out the possibility that *Wolbachia* may affect aggression in CR, GF and MB males, we generated *Wolbachia*-free flies using tetracycline for three generations as previous described (Hoffmann, et al. *Evolution*, 1986). We found that *Wolbachia*-free GF males still showed decreased aggression compared to *Wolbachia*-free CR males, and adding MB fully rescued aggression of *Wolbachia*-free GF males (see below figure, now added as supplementary Figure 7). These results clearly indicate that *Wolbachia* is not involved in the microbiome-mediated promotion of aggression. In fact, the previous study (Rohrscheib et al. *Appl. Environmental Microbiology*) showed that only one of three *Wolbachia* strains decreased male aggression, but the two other commensals had no effect on aggression, suggesting that only pathogenic *Wolbachia* negatively regulated aggressive behavior, distinct from the positive regulation of aggression by microbiome in our study.

2) At this stage, I still have some concerns regarding the specificity of the effect observed by the authors. Both at the level of behavior as well as mechanism. Therefore in my opinion it is important to look into the following points:

a) Is the change in aggression levels due to the flies now performing other behaviors instead of aggressive ones? I am mainly concerned about feeding behavior competing with the aggressive drive of the flies. We have shown that the microbiome and dietary yeast alter feeding behavior in flies. Could it be that the flies which show low aggression levels are spending most of their time feeding or performing other behaviors like grooming? Then the effect of the microbiome would not be on aggression directly but rather on other behaviors which would then compete with the display of aggressive behaviors. The authors should therefore characterize the behavioral effect of microbiome manipulations in the flies in which they score aggression to test if the effect is mainly on aggression.

We appreciate that the microbiome could alter feeding choice as previously found (Henriques et al, Nature Communications, 2020; Leitão-Gonçalves et al, PLOS Biology, 2017), and set out to examine feeding and grooming behaviors of CR, GF and MB flies. The flies we used for the 30-min aggression test are fed or satiated males, so they should eat very little during aggression assay. Indeed, they showed equally low level of feeding, compared to starved males, under feeding assays using food with blue dye for 30 min. We also scored the percentage of time males perform grooming behavior in our aggression assay and found indistinguishably low level of grooming behaviors in CR-CR, GF-GF and MB-MB paired males. These results clearly showed that the decreased aggression in GF males is not attributed to competitions with other exclusive behaviors, such as feeding and grooming. This new result is now added as supplementary Figure 10 shown below.

b) I am puzzled that changing octopamine biosynthesis alters aggression specifically. Octopamine has been shown to affect all kinds of behaviors. An open question to me is: Is the effect of the microbiome only affecting a subset of neurons altering aggression or general octopaminergic control of behavior like locomotion, flight, egg-laying, etc.? Maybe that is where the authors are going with the analysis of locomotion but they do not necessarily spell out clearly if their findings suggest that the microbiome manipulation only affects a subset of behaviors controlled by octopamine or all behaviors controlled by octopamine. If it is a specific effect on aggression how do the authors explain it? Does it mean that the effect of the microbiome during early development only affects specific neurons? If yes which and how? This can be tested experimentally and would be an important addition to the paper. This would require a more thorough analysis of the anatomical impact of microbiome manipulations than currently done in Figure 4 (it is difficult to see anything in the current brain stainings).

We thank the reviewer for this suggestion. To test whether microbiome affects Tdc2 expression in specific subsets of octopaminergic neurons, we divided Tdc2-expressing neurons into five distinct clusters based on previous classifications. We quantified the fluorescence intensity of anti-Tdc2 signals in these five subsets of neurons, and found that anti-Tdc2 signals in three of five subsets of Tdc2 neurons are decreased in GF males (see below figure, now as part of Figure 4). Consistent with the decreased anti-Tdc2 signals, we further monitored spontaneous Ca^{2+} signals in these subsets of Tdc2 neurons using GCamp6m, and found decreased Ca^{2+} signals in the same three subsets of Tdc2 neurons (now as Supplementary figure 11). These results indicate that only subsets of octopaminergic neurons were affected by depletion of microbiome.

While we could not explain why Tdc2 expression was decreased in only subsets of neurons (but not all Tdc2-expressing neurons), which certainly needs future investigation, we checked when this effect took place during development. We found that numbers of Tdc2-expressing neurons in neither 2nd instar larva (36~48 hours AEL) nor 3rd instar larva (72~84 hours AEL) were grossly affected by microbial depletion. However, Tdc2 expression was significantly decreased in subsets of Tdc2-expressing neurons of GF males during 3rd instar larval stage, but unaffected during 2nd instar larval stage, suggesting that the microbiome-mediated increase in octopamine signal originates approximately from the 3rd instar larval stage (see figure below, now as a part of Figure 5). These results indicate that specific subsets of octopaminergic neurons are particularly vulnerable to early-life manipulations of the microbiome.

Regarding to why microbiome only affects aggression but not locomotor or other behaviors, there are at least two possibilities: one is that the OA level is not significantly reduced in specific neurons that may be responsible for a particular behavior (e.g., neurons in the ventral nerve cord for egg-laying); and the other is that although OA is reduced in certain neurons, it is not sufficient to induce a behavioral change (e.g., even *tβh* or *tdc2* mutant flies showed comparable locomotor levels, although they are defect in starvation-induced hyperactivity, Yang et al., 2015, PNAS). These possibilities were added into the discussion part.

c) The authors suggest that the main effect of the microbiome is to alter Tdc2 levels. If they could show that they can rescue the effect of removing the microbiome on aggression, by

overexpressing Tdc2 acutely in the adult from a transgene in GF flies, that would show that no other developmental effect like changes in circuit architecture is at play.

We thank the reviewer for this suggestion. For this experiment, we need UAS-Tdc2 line from the Bloomington Drosophila Stock Center. Due to Covid-19, we unfortunately haven't received this stock yet. Although we could not do such experiment currently, we already showed that adding OA agonist CDM just two days before aggression assay during adulthood acutely restored aggression in GF males (see data below), which strongly excludes the possibility of developmental defects in the neural circuit level. We hope this addresses the reviewer's concern.

3) I am puzzled that the authors find no effects of adding heat-killed bacteria or the supernatant of the bacterial culture to the flies. Especially given that a possible explanation of the effect of the microbiome could be the increase in protein absorption in development. Obviously, heat-killed bacteria would also have proteins. These are not straight forward experiments as one has to make sure that one adds the same level of bacterial material as present in the fly at the stage when it is provided. One has to take into account that one provides much higher titers than when inoculating the flies with living bacteria as heat-killed bacteria do not grow exponentially. Did the authors do this? I could not find any description of how that experiment was done in the manuscript. I think it is missing in the materials and methods. The authors should therefore describe the experiment in the methods section and ensure that the level of heat-killed bacteria material is the same as presenting the food when flies are alive. This very likely means providing heat-killed material repeatedly throughout development and at high levels. This has been done by us or the Leulier lab in our most recent publications.

We thank the reviewer for this important suggestion and carried out heat-killed bacteria supplementation as described (Henriques et al., Nature Communications, 2020, and Storelli et al., Cell Metabolism, 2017). To ensure that bacterial load approximates the one in the gnotobiotic experiments, we first assessed the bacterial titres in the fly food vials. Bacteria reached a plateau at 2.6×10^7 CFUs/ml after 4 days of growth under this condition. We added the 10-fold amount of heat-killed bacteria (2.6×10^8 CFUs/ml) to GF flies. Our data showed that GF flies with high levels of dead bacterial biomass did not show an increase in aggression compared to their counterparts, suggesting that the bacteria must be metabolically active to promote aggression. These details were added into the materials and methods in the revised manuscript. This new result is now added as Supplementary figure 5 shown below.

Conceptually:

In my opinion, the finding that the effect of the microbiome on adult behavior results from a dietary interaction with the microbiome at an early developmental stage is exciting and intriguing. There is a very straight forward hypothesis which emerges from work in the field, including ours, and which the authors could easily test and would point to a clear mechanism by which the microbiome alters behavior. The most straight forward hypothesis is that the microbiome (specifically Lp) increases the uptake of amino acids from the diet which then alters octopamine biosynthesis in the adult. The authors can easily test this hypothesis by expanding the experiments presented in Figure 5e. By switching flies to diets with different levels of yeast or amino acids at the different time points after egg laying (same as the ones when they provide the microbes) they should be able to test if they can i) revert the effect of the microbiome and ii) mimics the effect of the microbiome. An increase beyond 10% of yeast during the 0-48h AEL time window should lead to changes in Tdc2 expression and aggression in GF animals. Conversely, a reduction to 0.5% or 2.5% of yeast when the animals are associated with the microbiome should abolish the effect of the microbiome on aggression. This could show that the effect of the microbiome is specifically to increase AA absorption in the juvenile leading to a change in neuromodulation.

We appreciate the reviewer for these valuable suggestions that could expedite our understanding how interactions of microbiome and nutrition affect aggression in *Drosophila*. We have examined effects of fly food with higher dead yeast (15% and 20%; fly food with more than 20% yeast is hard to prepare), but our results showed that autoclaved yeast in massive excess still failed to promote aggression to the extent of live bacteria, suggesting that live microbes are necessary to optimize host aggression. This is added as Supplementary Figure 13 shown below.

We next asked whether undernutrition during development abolishes the effect of the microbiome on aggression in CR males. We firstly reared CR embryos in food with 0.5% yeast, and switched them to diets with 10% yeast at different time points after egg laying (AEL). We found a significant decrease in aggression, as well as decreased *Tdc2* expression, in adult males if raised with 0.5% yeast for more than 2 days AEL, and the effect is stronger if raised with 0.5% yeast for longer time. These results indicate that malnutrition during a critical developmental period is sufficient to abolish the effect of the microbiome on aggression of males. This new result is now added as a part of Figure 6 shown below.

We also tested whether undernutrition after the critical developmental period would impair male aggression. We firstly reared CR embryos in food with 10% yeast, and switched them to diets with 0.5% yeast at different time points AEL. We found that males showed decreased aggression and *Tdc2* expression if transferred to 0.5% yeast food within 3 days AEL, but regular aggression and *Tdc2* expression if transferred to 0.5% yeast food 4 days AEL. Together these results reveal a critical developmental period (2-4 days AEL) when nutrition is necessary to sustain microbiome-mediated aggression of males. This new result is now added as a part of Figure 6 shown below.

Also, the authors should test if a continuous association with the microbiome is required to give an effect on aggression. What happens if the flies are manipulated to lose their microbiome at 24h, 36h, and 48h after being associated with microbes at 36h AEL. These straight forward experiments should be able to clarify if the microbiome mainly acts on adult aggression by altering AA uptake during development and what is the exact time window when the microbiome acts.

We appreciate the reviewer for this important experiment. We raised CR embryos with regular food and transferred them at different time points onto food with a cocktail of antibiotics. We found that depleting bacteria within 1-3 days AEL dramatically reduced male aggression and Tdc2 expression levels, while depleting bacteria 4 days AEL did not affect aggression or Tdc2 expression of adult males. These results further indicate that both microbiome and nutrition are required during a critical developmental period (2-4 days AEL) for the manifestation of aggression in adult males. This new result is now added as a part of Figure 5 shown below.

Finally, I would like to close by encouraging the authors to expand their introduction and discussion to include aspects which I think expand the impact of their paper by making it more interesting to a wider audience. There is a growing recognition that the effect of the microbiome on brain function and behavior can be best understood in the context of the interaction of the microbiome with diet. The current work makes this case very nicely too and I think that the authors could profit from placing their paper in this context more explicitly. This is a core finding and makes total sense in the context of how many people think about

the problem in the field. For relevant literature, they can look at work cited in reviews of John Cryan (e.g. Sandhu et al. Transl. Research 2017) and our recent review (Ezra-Nevo et al. Current Opinion in Neurobiology 2020).

I also think that there are two important big questions which the authors hardly discuss but which are very pertinent given their findings. First, it would be interesting if the authors could discuss how the microbiome could act on diet to change octopamine signaling, how these changes could be propagated to the adult stage, and most importantly why this is the case. Are the authors looking at an adaptive mechanism or is this a pathological situation? If I might venture to propose an idea: One could envisage that animals exposed to a rich diet are more likely to also be better mates than animals exposed to malnutrition. Therefore it would make sense to have them be more aggressive and hence dominant and more likely to pass on their offspring. But here we are in the realm of speculation. This is just an idea for the authors to interpret their data, which I do not expect them at all to pick up.

We thank the reviewer for these valuable suggestions and detailed references. We have now expanded our introduction and discussion in the context of the interaction of the microbiome with diet during a critical developmental period. All textual changes are in track change mode in the revised manuscript.

To finalize I think this is an exciting manuscript with interesting discoveries. I congratulate the authors for their work in such difficult times. Some technical issues remain which could severely alter the validity and interpretation of the data but they should be straight forward to address.

We thank the reviewer for so many constructive suggestions, and we have done most, if not all, suggested experiments, and we believe this version of manuscript has been much improved.

Reviewer #2 (Remarks to the Author)

This is a revised manuscript by Jia et al., describing the impact of the gut microbiome on *Drosophila* aggression and on the production of octopamine. The topic remains significant and the results will be of interest to the gut-brain community as a whole. The language and writing of the paper as a whole has improved with more information on rationale and less comparisons with experiments from other groups.

Many of the major and minor points were addressed in this revision. Several remain which need to be need to be attended to before I can recommend publication.

Major points.

1. To demonstrate that germfree flies were generated, the authors now show 16S rDNA PCR results and growth on nutrient agar plates. The nutrient agar plates in Fig. 1B are too small and are not labeled. This can easily be fixed or enlarged and moved to a supplemental figure. 16S rDNA PCR results are shown for the Canton S wild type and GF lines – but not for Tdc2-Gal4, UAS-TrpA1, or UAS-NaChBac. In my previous review, I wrote to the authors “it is critical to show in a supplemental figure that each line is GF”. The PCR experiments were not provided to demonstrate that the Gal4 line and UAS lines or alternatively the progeny

were GF and they need to be, this is a fundamental part of science.

We thank the reviewer for these valuable suggestions. The 16S rDNA PCR results and growth on nutrient agar plates are now moved to be Supplementary Figure 1 as suggested. We have testified axenia of each GF lines using 16S rDNA PCR. This new result is now added as Supplementary Figure 12 shown below.

2. Additional information on the aggression assay was provided but this information is not satisfactory. The authors write that the lunge number was manually counted for 30 minutes. I wrote in my previous review “The authors need to show the lunge number data even if it is a supplemental figure.” This was not done. In the rebuttal letter, the authors state, “we put all data into a source data files as a supplemental file now. I have the supplemental excel files (which is easy to follow and clear) in front of me. The 30 min lunge number is not provided, still only 10 minutes. The authors ran the assay for 30 mins, quantified all lunges within 30 mins as the materials and methods say, and I simply requested that the lunge number for all 30 mins be placed in a supplemental figure. This does not require any new experiments, gives an accurate picture of the fighting behavior during the fight assay which is important due to previous reports and could highlight new interesting findings.

We appreciate the reviewer for these suggestions. In our experiments, we recorded videos of aggression for 30 min, and we first analyzed how long it took (latency) for males to initiate lunges, in most cases, within 5 minutes. Thus, we could analyze aggression with maximally ~25 minutes. We previously only analyze aggression for 10 minutes as it is very time-consuming for manual score of the lunging behavior. To address the reviewer’s concern, we re-analyzed lunging behavior in CR, GF and MB males until the end of the video (see figure below), with each lunge being marked.

We then analyzed the lunge numbers within 20 min after fighting initiation, as well as lunge numbers in the first 10-min and second 10-min, and revealed that GF males have reduced aggression in both the first 10-min and second 10-min periods (and of course the total 20 min analyzed). Note that lunge numbers in the first 10-min are generally more than those in the second 10-min in all males, suggesting a decrease of aggression over time, regardless of CR, GF or MB males. Thus, lunge numbers in the first 10-min period well represent levels of aggression in males. We hope this new analysis addressed the reviewer's concern. We now updated the source data and added the above figure as Supplementary Figure 3.

Related: please add the median line to scatterplots on all figures.

We replaced the previously thin dotted median line with a wider solid line in all new figures.

3. The ROI interest data does not make sense in its current form. The authors need to put the ROI box around smaller areas. For ROI-1, draw a box around a pair of neurons on one side, a single neuron, or a very small cluster of neurons and redo the quantification. It is apparent there are neurons at the top of the box that are expressing Tdc2 weakly. If the authors are trying to emphasize this area, then draw a box there but not what is currently used. The same should be applied to ROI-2.

Thanks for the valuable suggestion. We have now re-analyzed this data. We divided the anti-Tdc2 signals into five smaller ROIs based on previous classifications, and quantified the fluorescence intensity of each ROI. We found that anti-Tdc2 signals in three of five ROIs were significantly decreased in GF males. This new result is now added as a part of Figure 4 shown below.

4. The Tdc2-Kir2.1 experiments were previously done by Hoyer 2008 in CR control males. Therefore, this experiment is not supportive in this situation. If MB male aggression was different than CR males this experiment would be useful. This data could be removed or if kept, then make sure to describe in the text that this experiment and reference the Hoyer paper.

We now removed the Tdc2-Kir2.1 data as suggested.

Minor:

Line 112, lunges not “lungs”

Corrected.

Reviewer #4

I am assigned as an additional reviewer to evaluate the revised manuscript, especially based on whether the reviewer 3's concerns were addressed appropriately. In this revised manuscript, the authors seemed to perform many experiments to support their claim and revise the text accordingly. In my point of view, the authors successfully addressed many concerns raised during the initial review. 16S data was nicely added to discuss the microbiome of their fly strains. Dietary manipulations increased the generality and robustness of the idea behind the early-life environment vs adult behaviour. Methods and legends are now firmly described.

The points which was not addressed in the revised manuscript;

1, To discuss the discrepancy, it would be better to use Oregon R to see locomotion. This is pointed out by R3 and other reviewers but still not included.

We thank the reviewer for this suggestion. We carried out locomotion with Canton-S and Oregon-R flies, and found that they have comparable locomotor activity. However, Oregon-R males do have significantly lower level of aggression. That is why we only carried out aggression assays using Canton-S males in this study. These results also indicate that aggression is not correlated with locomotor behaviors, consistent with our findings that GF males have reduced aggression but regular locomotor activity. This new result is now added as Supplementary Figure 9 shown below.

2, Using *L. plantarum* as a “representative gut microbiota” is not well justified. They mentioned their flies contain Lactobacillaceae as a dominant family, and that *L. plantarum* is a dominant bacterial species. This seems not the case as the majority of gut microbiome is from Acetobacteraceae (Fig. S3). I would like to recommend Acetobacter/Commensalibacter isolated from their flies to test the behaviour. Otherwise, *L. brevis* should have been used as pointed by R3. Although this is not mandatory, it is slightly disappointing to lose the opportunity to discuss species specificity/generality, and to discuss the above-mentioned discrepancy.

We thank the reviewer for these valuable suggestions. Indeed, *L. plantarum* is not the richest bacterial species in fly guts according to our sequencing data; however, *L. plantarum* is abundant and globally exists in lab-reared and wild-captured *Drosophila* (Chandler et al., PLoS Genetics, 2011), although components of *Drosophila* microbiome are temporally and spatially dynamic and diverse. Therefore, *L. plantarum* was frequently selected as a representative gut microbiome by many previous studies (Henriques et al., Nature Communications, 2020; Storelli et al., Cell Metabolism, 2017; Consuegra et al., PLoS Biology, 2020). In addition to use *L. plantarum*, we now tried to add some other bacterial species that we have isolated and identified in fly guts to GF embryos, and found that commensal bacteria including *Acetobacter*, *Lactobacilli* and *Enterococci* promote aggressive behavior just like *L. plantarum*. In contrast, promotion of aggression was not observed when a non-commensal microbe (ECC15 or a fungus) was used. These results indicate that a wide range of commensal bacteria could promote aggression, probably through similar mechanisms of elevating OA production. This new result is now added as Supplementary Figure 7 shown below.

Reviewers' comments:

Reviewer #1 (Remarks to the Author):

I would like to congratulate the authors on the excellent revision of the manuscript. They satisfactorily address all my concerns and I think strongly strengthened the paper both in terms of insights as well as opening fascinating future paths of inquiry. I think that especially the narrowing down of the critical window is spectacular. This has shaped up to be a fantastic paper.

I only have one major issue with a new set of data and its interpretation. In the new supplemental figure 11, the authors claim to survey the activity of the Tdc2 neurons and to be able to recapitulate the difference in Tdc2 levels at the level of activity. This is however not what the authors are showing. The authors are looking at absolute levels of GCaMP signal which without calibration does not allow them to make statements about the activity of these neurons. They would need to correct for baseline differences in expression using another reporter of baseline expression ($\Delta F/F_0$). Also, in this case, it is important to note that the authors use Tdc2-Gal4 to drive the expression of the GCaMP reporter. What the authors are therefore reporting is changes in expression in GCaMP which is due to changes in Tdc2-Gal4 activity induced by the microbiome manipulations. This nicely supports their conclusions using a transcriptional reporter but does not allow them to draw any conclusion at the level of the activity of the neurons. In the current form, these data should be removed. Importantly, this does not in any way weaken the conclusions of the authors or precludes the publication of the manuscript.

Minor comments:

I personally think that the authors overuse the word “demonstrate” or “unambiguously demonstrated” (lines 286, 310, 321). The data are nice and convincing but I can come up with many models which are compatible with the results and do not fit the conclusion of the authors. These are however all much less likely. I would suggest the use of strongly indicate or something similar.

I am also not sure why the authors conclude that the critical time window is 48-96. I would have thought 0-96h is the more likely window.

Carlos Ribeiro

Reviewer #2 (Remarks to the Author):

I have read the second revision of the manuscript by Jia et al., describing the impact of the gut microbiome on *Drosophila* aggression and on the production of octopamine.

My concerns have been addressed and I congratulate the authors on putting together an exciting story.

Reviewer #4 (Remarks to the Author):

The authors of the study addressed my concerns thoroughly. The revised manuscript is now improved and therefore I can support the publication.

Our point-to-point responses are in blue.

Reviewer #1 (Remarks to the Author):

I would like to congratulate the authors on the excellent revision of the manuscript. They satisfactorily address all my concerns and I think strongly strengthened the paper both in terms of insights as well as opening fascinating future paths of inquiry. I think that especially the narrowing down of the critical window is spectacular. This has shaped up to be a fantastic paper.

We thank the reviewer for this very positive comment on our manuscript, especially many constructive comments during the whole review process, which helped us to improve the manuscript.

I only have one major issue with a new set of data and its interpretation. In the new supplemental figure 11, the authors claim to survey the activity of the Tdc2 neurons and to be able to recapitulate the difference in Tdc2 levels at the level of activity. This is however not what the authors are showing. The authors are looking at absolute levels of GCaMP signal which without calibration does not allow them to make statements about the activity of these neurons. They would need to correct for baseline differences in expression using another reporter of baseline expression ($\Delta F/F_0$). Also, in this case, it is important to note that the authors use Tdc2-Gal4 to drive the expression of the GCaMP reporter. What the authors are therefore reporting is changes in expression in GCaMP which is due to changes in Tdc2-Gal4 activity induced by the microbiome manipulations. This nicely supports their conclusions using a transcriptional reporter but does not allow them to draw any conclusion at the level of the activity of the neurons. In the current form, these data should be removed. Importantly, this does not in any way weaken the conclusions of the authors or precludes the publication of the manuscript.

We agree with the reviewer on the limit of this supplemental figure, and we would like to remove this data as suggested by the reviewer. We also made textual changes accordingly.

Minor comments:

I personally think that the authors overuse the word “demonstrate” or “unambiguously demonstrated” (lines 286, 310, 321). The data are nice and convincing but I can come up with many models which are compatible with the results and do not fit the conclusion of the authors. These are however all much less likely. I would suggest the use of strongly indicate or something similar.

We have made such textual changes throughout the manuscript whenever necessary.

I am also not sure why the authors conclude that the critical time window is 48-96. I would have thought 0-96h is the more likely window.

Our data showed that bacteria recolonization within 48h AEL (*e.g.*, 36h AEL) fully restored aggression level of GF males (Figure 5b), which indicates that microbiome is not required during these early stages for adult aggression. This is why we concluded that the critical time window is 48-96h instead of 0-96h.

Reviewer #2 (Remarks to the Author):

I have read the second revision of the manuscript by Jia et al., describing the impact of the gut microbiome on *Drosophila* aggression and on the production of octopamine.

My concerns have been addressed and I congratulate the authors on putting together an exciting story.

We thank the reviewer for the constructive comments during the whole review process.

Reviewer #4 (Remarks to the Author):

The authors of the study addressed my concerns thoroughly. The revised manuscript is now improved and therefore I can support the publication.

We thank the reviewer for supporting publication of this paper.